# Advanced Glycation End-Products (AGEs): Formation, Chemistry, Classification, Receptors, and Diseases Related to AGEs

**DOI:** 10.3390/cells11081312

**Published:** 2022-04-12

**Authors:** Aleksandra Twarda-Clapa, Aleksandra Olczak, Aneta M. Białkowska, Maria Koziołkiewicz

**Affiliations:** Institute of Molecular and Industrial Biotechnology, Lodz University of Technology, Stefanowskiego 2/22, 90-537 Lodz, Poland; aleksandra.olczak@dokt.p.lodz.pl (A.O.); aneta.bialkowska@p.lodz.pl (A.M.B.); maria.koziolkiewicz@p.lodz.pl (M.K.)

**Keywords:** advanced glycation end-products, AGEs, AGE formation, AGE classification, AGE receptors, RAGE, Stab2, AGE-related diseases

## Abstract

Advanced glycation end-products (AGEs) constitute a non-homogenous, chemically diverse group of compounds formed either exogeneously or endogeneously on the course of various pathways in the human body. In general, they are formed non-enzymatically by condensation between carbonyl groups of reducing sugars and free amine groups of nucleic acids, proteins, or lipids, followed by further rearrangements yielding stable, irreversible end-products. In the last decades, AGEs have aroused the interest of the scientific community due to the increasing evidence of their involvement in many pathophysiological processes and diseases, such as diabetes, cancer, cardiovascular, neurodegenerative diseases, and even infection with the SARS-CoV-2 virus. They are recognized by several cellular receptors and trigger many signaling pathways related to inflammation and oxidative stress. Despite many experimental research outcomes published recently, the complexity of their engagement in human physiology and pathophysiological states requires further elucidation. This review focuses on the receptors of AGEs, especially on the structural aspects of receptor–ligand interaction, and the diseases in which AGEs are involved. It also aims to present AGE classification in subgroups and to describe the basic processes leading to both exogeneous and endogeneous AGE formation.

## 1. Advanced Glycation End-Products (AGEs)—Introduction, Chemistry, Classification

AGEs belong to a heterogeneous, complex group of compounds formed either exogenously or endogenously by different mechanisms and from a variety of precursors. In general, AGEs are created by means of non-enzymatic condensation (without any biological catalyst involved) between carbonyl groups of reducing sugars and free amine groups of nucleic acids, proteins, or lipids, followed by further rearrangements yielding stable, irreversible end-products.

Reactions leading to the formation of AGEs have been known for more than a hundred years, as their description by Maillard dates back to 1912 [1]. He investigated the non-enzymatic browning observed in the processes of baking or broiling in the reaction of glucose with glycine. Further discoveries in the area of food chemistry have identified the end-products of the Maillard reaction as the compounds giving a richness in taste, aroma, and savor to thermally processed foods with high sugar and protein content. The developing western diet has increased the consumption of highly processed foods. Chemical compounds are often added to enrich food taste, texture, or safety, but they could also undergo reactions leading to AGE formation upon thermal processing. Food and drinks have become packed with sugars and fats; protein supplementation is also used to enhance muscle buildup. In parallel with decreased physical activity, the western lifestyle has led to civilization diseases and has brought the attention to research on the etiology of pathologic states, such as insulin resistance, diabetes, cardiovascular diseases (CVD), cancer, and aging. Some attention should be paid to the in vivo glycation of proteins. In the human body, AGE formation is a prolonged process. The first observed endogenous early glycation product, glycated hemoglobin (HbA1c), was detected in 1968 in a human body in a state of diabetes [2]. This was followed by further discoveries of AGEs in different health conditions [3]. For several decades, AGEs have been implicated in many signaling pathways leading to inflammation states and related diseases. The number of publications reporting experimental data on both endogenous and exogenous AGEs is rapidly growing, indicating a growth in evidence of their involvement in many physiological and pathological processes [4,5,6].

This review focuses on the receptors of AGEs, especially on the structural aspects of receptor–ligand interactions, and the diseases in which AGEs are involved. It also aims to present AGE classification in subgroups and to describe the basic processes leading to the formation of this heterogeneous group of glycated proteins.

### 1.1. Formation and Chemistry of AGEs

The diverse nature of the structures of AGEs suggests a great variety in the mechanisms of their formation. Indeed, both endogenous and exogenous AGEs can be generated in several pathways from a number of precursors (Figure 1).

The first step of the Maillard pathway consists of the condensation of a carbonyl group, e.g., from a reducing sugar and an amine group from moieties such as the side chains of lysine or arginine [1,7]. Figure 1 represents a scheme of the generation of AGEs starting from a reducing aldose or a reducing ketose. Reducing ketoses can be also produced in vivo by way of the polyol pathway: D-glucose is reduced to sorbitol by aldose reductase and further transformed by sorbitol dehydrogenase to D-fructose. Condensation of the carbonyl group and the amine group is a reversible reaction yielding the so-called Schiff base (aldimin). In the Maillard pathway, it undergoes Amadori rearrangement, giving a stable Amadori product. Its analogue in the polyol pathway is the Heyns product created by Heyns rearrangement [8]. These compounds are considered early glycation products and include, e.g., fructose-derived AGE (Fru-AGE) and glucose-derived AGE (Glu-AGE; Figure 1) [9]. HbA1c and glycated human serum albumin (sometimes abbreviated as gHSA or GHSA) [10] belong to the latter class and are important markers of hyperglycemic states, insulin resistance, and diabetes [11,12,13,14].

It has been demonstrated both in thermally processed food and in vivo that different sugars vary in susceptibility to the Maillard reaction [15,16]. The cyclic forms of sugars are thermodynamically favored and most abundant, but their open chain forms, ranging from 0.0002% for D-glucose to 0.7% for D-fructose, are the reactive isoforms of monosaccharides undergoing non-enzymatic glycation [17]. This is why D-glucose exhibiting high stability in its pyranose form is a less reactive sugar, whereas D-fructose with more abundant chain isoforms is about 7.5-fold more reactive [18,19].

Another factor with a significant impact on the formation of early glycation products is temperature. In the case of food processing at high temperatures (starting from c.a. 120–180 °C), the Maillard reaction is very rapid, while the in vivo formation of Amadori and Heyns products requires much longer time [20].

To become AGEs, the end-compounds created during the Maillard and polyol pathways, along with their intermediates (e.g., Schiff bases), must undergo further rearrangements, producing the final AGEs and a heterogeneous class of reactive carbonyls. Several of the most-recognized examples include glyoxal, methylglyoxal, glyceraldehyde, glycolaldehyde, diacetyl, and 1- and 3-deoxyglucosone (Figure 1). Generation of AGEs by the autoxidation of Amadori products is called the Hodge pathway [20]. Other routes are the Namiki pathway (amino acid or lipid degradation and cleavage of dicarbonyl compounds from aldimins) and the Wolff pathway, constituting the formation of carbonyls after autoxidation of monosaccharaides (glucose, fructose, ribose, and glyceraldehyde) [21].

Reactive carbonyls can be also originated from other various pathways in vivo, e.g., methylglyoxal, and glyceraldehyde can be generated in the course of glycolysis (from glyceraldehyde 3-phosphate) or fructolysis (from fructose 1-phosphate). Other instances leading to the formation of this class of compounds are glucose oxidation and the metabolism of D-fructose yielding triose phosphate and fructose 3-phosphate. At this point, it should be also noted that another important group of compounds, advanced lipoxidation end-products (ALEs), originates from the reactive carbonyls formed as a result of lipid peroxidation [7]. However, although AGEs and ALEs may share the same structure because of formation from the same precursors, e.g., glyoxal or methylglyoxal, they are not reviewed in this paper, as they are two separate groups.

The reactive carbonyls can undergo further condensation with the available amine groups from lysines and arginines, as in the case of the Maillard reaction. These reactions yield a great variety of final AGEs, which generally may be divided into protein adducts and protein crosslinks. The AGEs derived from different molecules can be termed with the prefix indicating their origin, e.g., glyoxal-derived (GO-AGEs), methylglyoxal-derived (MGO-AGEs), glyceraldehyde-derived (Glycer-AGEs), glycolaldehyde-derived (Glycol-AGEs), 3-deoxyglucosone-derived (3-DG-AGEs), etc. Certain AGEs (such as N^ε^-(carboxymethyl)lysine, CML; or 6-{1-[(5S)-5-ammonio-6-oxido- 6-oxohexyl]imidazolium-3-yl}-L-norleucine/glyoxyl-derived lysine dimer, GOLD; Figure 1) may be created from more than one precursor. The processes leading to the generation of AGEs have been described in fine detail in several review articles [7,20].

### 1.2. Types of AGEs

Complex mechanisms of AGE formation, as well as a great variety of precursors, generate many chemically diverse AGE molecules that have so far been identified in human blood, tissues, and in foods [7,22,23]. Their chemical structures, properties, origins, and physiological significance allow their categorization into several groups when certain criteria are considered (Table 1). The chemical structures of the chosen model compounds are presented in Figure 2.

#### 1.2.1. Endogenous and Exogenous AGEs

The first criterion defining an AGE refers to origin, dividing AGEs into compounds absorbed from food (exogenous and dietary, dAGEs) or generated inside an organism. The latter, endogenous AGEs, are usually formed in all tissues and body fluids under physiological conditions in the reactions of glycation, comprising an important element of normal sugar metabolism in the cell [20]. Endogenous AGEs can be further divided with respect to their precursor (Table 1). GO-derived compounds include GOLD, GODIC, CMA, and CML; MGO-derived include MG-H1, MODIC, MOLD, argpyrimidine, and CEL; and 3-DG-derived include pyrraline, 3DG-H1, pentosidine, DOLD, and DOPDIC. The full chemical names and structures of these compounds are presented in the legend of Figure 2.

In a cell, the excessive accumulation of AGEs may have impacts on both long- and short-lived proteins and peptides [7]. In the case of long-lived proteins and peptides, these compounds become pathogenic. This often refers to elderly people, especially patients with diabetes mellitus with elevated glucose concentrations generating excessive AGE accumulation. In healthy subjects, the levels of glycated products of plasma proteins are less than 3% [24]. On the other hand, in pathological situations such as diabetes, this level can increase up to three-fold, resulting in the development of a pathological phenotype [25,26]. The pathologic effects of AGEs are related to their ability to promote oxidative stress and inflammation by binding with cell surface receptors or crosslinking with body proteins and altering their structure and function [27]. AGEs, through oxidative stress, lead to the activation of several stress-induced transcription factors with the production of proinflammatory and inflammatory mediators, such as cytokines and acute-phase proteins [22]. Taken together, these events lead to the development of chronic diseases, including CVD and diabetes. Moreover, AGEs are responsible for progression of musculoskeletal diseases such as osteoarthritis, which is nowadays a common disorder leading to disabilities in elderly people. A high level of endogenous AGEs may also have an impact on the development of rheumatoid arthritis [28]. In this case, the modified compounds increase the stiffness of collagen networks in the bone, which leads to increased skeletal fragility and risk of bone fractures [29].

Concerning the metabolism of endogenous AGEs, it is thought that these compounds are not substrates for enzyme systems involved in detoxification by phase 1 and 2 enzymes; they are probably metabolized by innate defense or intracellular degradation after receptor-dependent uptake [20].

It should be also noted that the abnormal accumulation of AGEs does not necessarily need to be dependent on the age and physiological state of cells, since these compounds are abundant in exogenous sources, such as food and tobacco smoke [27,30,31]. AGEs are present naturally in uncooked animal-derived foods, but their amount increases with the method of processing (especially thermal). Grilling, roasting, broiling, searing, and frying are particularly favorable for AGE formation. These processes are often indispensable stages of preparation for modern food products, as they increase palatability (e.g., enhance flavor, color, and appearance), prolong shelf-life, and reduce food-borne diseases [27,31]. Most often, however, such processed food in modern diets (e.g., dry mixes or canned soups) are rich in sugars and proteins that, during thermal processing, undergo the Maillard reaction, leading to rapid AGE formation. It was initially thought that dAGEs were poorly absorbed by cells, and their impact on human health was ignored to a large extent. This hypothesis was, however, quickly combatted by research on mice fed with food supplemented with specific AGEs [32,33,34], which was followed by the detection of a significant pool of these compounds in the cells.

Stable and relatively inert CML, one of the most well-described AGEs, was used as a representative for the determination of AGE levels in chosen unprocessed and processed food products [31]. Table 2 presents the amounts of CML from the enzyme-linked immunosorbent assay (ELISA) in AGE kU/100 g. Another database presenting an even greater diversity and amount of dAGEs was created by the Technical University of Dresden, Faculty of Science, Department of Food Chemistry (AGE Database) [35]. This source refers to the amounts of AGEs in chosen food products given in mg per 100 g of protein and defines the analytical method used to quantify these compounds, which is supported by the literature data. Generally, meats contain the highest AGE levels per standard serving size, whereas vegetables, fruits, and whole grains are low in AGEs (Table 2) [31,36].

#### 1.2.2. Fluorescence and Crosslinking of AGEs

Considering the diversity of chemical structures and the ability to emit fluorescence, Perrone et al. proposed another division of AGEs [22]: non-fluorescent and crosslinked compounds including, e.g., GOLD, MOLD, DOLD, and MODIC; fluorescent and crosslinked including pentosidine and crossline; fluorescent and non-crosslinked including argpyrimidine; and, finally, non-fluorescent and non-crosslinked including CEL, CML, and pyrraline (Figure 2). Among these, so far, the best-characterized are pentosidine and CML [37]. The latter has often been used as a surrogate for representative AGE molecules. There are many known methods of CML determination [31,38,39,40]. However, it should be noted that, in a diet, especially based on highly processed foods, many AGEs different than CML are present and are also associated with the severity of diseases [41]. Further methods for the determination of AGE levels, whether or not they rely on intrinsic fluorescence, will surely be developed as novel biomarkers [42,43].

#### 1.2.3. Low-Molecular-Weight AGEs (LMW) and High-Molecular-Weight (HMW) AGEs

AGEs can be classified based on their molecular weight into LMW-AGEs and HMW-AGEs. Gerdemann et al. classified LMW-AGEs as proteins whose mass does not exceed 12 kDa; the remaining proteins are grouped as HMW-AGEs [44]. It is thought that the latter are considered to be protein-bound proteins, while LMW-AGEs are free proteins or peptide-bound molecules [45,46]. Significantly more information can be found in the literature on the topic of LMW-AGEs, which comes from their simpler structure and higher stability. Thus, the analytical methods of quantification used for their detection are simpler and faster. It should be noted that LMW-AGEs, especially during heat treatment, can aggregate and form HMW-AGEs [47].

The creation of LMW-AGEs can be a result of incomplete degradation of proteins. Compounds such as CML can be absorbed via simple diffusion, while peptide-bound AGEs use peptide transporters to cross the epithelium [48]. This means that LMW-AGEs are readily distributed throughout the body. Their accumulation can be observed mainly in the liver, kidneys, lungs, heart, and spleen. It was hypothesized that the distributions of LMW- and HMW-AGEs throughout the body is different, and their fate can be dependent on factors such as exact structure, host diet, and gut environment [47].

#### 1.2.4. Non-Toxic and Toxic AGEs (TAGEs)

Takeuchi et al. recently proposed the division of AGEs into non-toxic AGEs and TAGEs [49]. This concept assumes that some reactions of the in vivo formation of AGEs have a physiological role as they induce the post-translational modification of proteins or even a protective role in the case of excess aldehyde/carbonyl compounds caught and detoxified by proteins. These are the functions assigned to so-called non-toxic AGEs, such as CML, CEL, pentosidine, pyrraline, MG-H1, MOLD, and GOLD. These glycation products are mostly generated through an averting path. For instance, Ahmed et al., who first identified CML, suggested that the process responsible for the generation of non-toxic AGEs might be involved in the detoxification of most end-products of glycation/carbonyl stress in human body [50]. On contrary, a subset of AGEs triggering signaling after binding to the receptor for AGEs (RAGE) and inducing the production of reactive oxygen species (ROS), is strongly cytotoxic and classified as TAGEs [49].

Takeuchi et al. pointed at a specific group of TAGEs, Glycer-AGEs, which originate from glyceraldehyde, a triose sugar intermediate of glucose or fructose metabolism [49]. Glyceraldehyde is generated in cells, especially in hepatocytes, through three pathways: (i) glucose is converted to glyceraldehyde 3-phosphate during glycolysis and then non-enzymatically dephosphorylated; (ii) fructose is converted to glyceraldehyde during fructolysis by fructokinase and aldolase B; and (iii) in the state of hyperglycemia, glucose is converted to fructose in the polyol pathway, where glyceraldehyde formation is modulated by aldose reductase and sorbitol dehydrogenase. Many publications from the Takeuchi group [49,51,52,53] have reported a remarkable interaction between the intracellular levels of TAGEs and the development of pathophysiological processes associated with lifestyle-related diseases (LSRDs). Examples include non-alcoholic fatty liver disease (NAFLD), non-alcoholic steatohepatitis (NASH), CVD, infertility, Alzheimer’s disease (AD), arteriosclerosis, and cancer. Moreover, the excessive cellular accumulation of TAGEs can lead to cell damage or death, as in the cases of hepatocellular damage and pancreatic ductal epithelial cell damage, cardiomyocyte pulsation arrest and cell death, and myoblast cell death. In addition, TAGEs accumulated in cells leak into the blood, increasing their levels of circulation [49].

A new concept of TAGE formation proposed by Takeuchi et al. established a strong connection between the amount of Glycer-AGEs in cells and human nutrition habits [49]. A crucial role was ascribed to high quantities of sugar-sweetened beverages (SSB), especially sucrose and high-fructose corn syrup (HFCS), whose consumption altered sugar metabolism and increased in vivo TAGE levels. This is one of the reasons why the World Health Organization (WHO) in 2016 recommended the reduction of daily sugar intake to <5% of total energy intake by adults and children [54]. Sugar as a food ingredient has been associated with the pathogenesis of metabolic syndrome-related diseases, such as diabetes mellitus, CVD, NAFLD, and NASH, as well as with the aging process, which is accelerated by glycation-related protein damage.

Not only endogenous, but also exogenous, AGEs have a significant impact on TAGE accumulation. The intake of dAGEs depends mostly on the type of food and its processing: solid fats, fatty meats, full-fat dairy products, and highly processed foods (mainly fried, deep-fat-fried, roasted, and grilled dishes) are particularly rich AGE sources. Takeuchi and coworkers reported that, in normal rats whose diet contained AGE-rich beverages, increased expression of RAGE, vascular endothelial growth factor (VEGF) in the liver, and abnormal accumulation of TAGEs in hepatocytes were observed. The authors concluded that the interactions between extracellular TAGEs and RAGE that can modulate intracellular signaling, modulate the expression of chosen genes, induce ROS production in several cell types, and lead to the liberation of pro-inflammatory molecules may cause the pathological changes observed in LSRDs [49].

Takeuchi postulated new strategies leading to the inhibition of TAGEs in the cells: (i) eating meals with low dAGE levels (mainly steamed and boiled dishes), (ii) consuming food rich in insoluble and indigestible dietary fiber, which has the ability to absorb exogenous AGEs; (iii) eating meals in an appropriate sequence: a vegetables-first and carbohydrates-last meal sequence; (iv) and physical activity. Obeying these rules is a method of prevention of LSRDs such as CVD or diabetes [49].

## 2. AGE Receptors

The ingestion of exogenous AGEs and the formation of endogenous AGEs causes an activation of various signaling pathways initiated by a series of receptors (Table 3) [20,55]. The most investigated human receptor is the type I cell surface RAGE belonging to the immunoglobulin (Ig) superfamily. Several other receptors for AGEs have also been identified and include the AGE-R1, -R2, and -R3 receptors and a group of scavenger receptors, among others, stabilin-1 and stabilin-2 (Stab1 and Stab2, respectively). The expression profile of the listed types of receptors, their ligand binding profiles, available structural information, and their contributions to the triggering of various signaling pathways are described in the paragraphs below.

### 2.1. RAGE

Since the first RAGE purification by Neeper et al. from bovine lung endothelial cells [56], this single-pass type I transmembrane protein is now regarded as a representative AGE receptor on endothelium. Human RAGE encoded by the AGER gene, which has a full length of 404 amino acids, is composed of three extracellular domains, Ig-like V-type (V, amino acids 23–116), Ig-like C2-type 1 (C1, amino acids 124–221), Ig-like C2-type 2 (C2, amino acids 227–317), as well as a transmembrane helical fragment (amino acids 343–363), and a disordered cytoplasmic domain (amino acids 364–404) (Figure 3a). Ten isoforms produced by alternative splicing exist [64]. The receptor can be also found as a secreted soluble isoform (sRAGE) produced from truncated variants [65]. Contrary to scavenger receptors, the recognition of AGEs by RAGE is not followed by ligand endocytosis but leads to intracellular signal transduction, including the activation of transcription factor NF-κB [66,67].

Concerning structural information, several experimental models created by NMR or X-ray crystallography are available from the Protein Data Bank (PDB). The first two crystal structures of a VC1 fragment of RAGE were published in 2010 by Koch et al. and Park et al. (Table 4) and provide insight into a possible binding mode for AGEs [68,69]. The broader image of the structure covering the C2 domain was provided in 2013 by Yatime and Andersen [70]. In the same year, the first structure with a ligand (oligonucleotide) was solved by Sirois et al. The model of a complex with human S100-A6 was delivered three years later by Yatime et al. [71,72]. No X-ray structures with an AGE ligand are available so far; however, Kozlyuk et al. reported several complexes of VC1 domains with RAGE inhibitors (fragments and more complex compounds) (Table 4) [73]. In 2011, Xue et al. published a solution (NMR) structure of a CEL-containing peptide in a complex with the V domain of RAGE, which revealed that carboxyethyl moiety binds into the positively charged cavity of the V domain (PDB ID 2L7U) [74].

The A chains (chain N in the case of PDB ID 3O3U) of the available RAGE X-ray models generally present a consistent Ig-like fold; the A chains align to the PDB ID 3CJJ A chain (VC1 fragment) with low RMSD values (Figure 3b). RAGE inhibitors (except 3-(3-{[3-(4-carboxyphenoxy)phenyl]methoxy}phenyl)-1H-indole-2-carboxylic acid from 6XQ3) generally bind to distant sites from the DNA and AGE or S100-A6 clefts (Figure 3c) [73]. Analyses of the available structures suggested a mechanism of the ligand binding: they revealed the presence of a large, positively charged patch on the RAGE surface (colored blue in Figure 3d). This pocket is lined by lysines 37, 39, 43, 44, 52, 107, and 110 and arginines 48, 77, 98, 104, 114, 116, and 216, which are situated mostly in the V, but also partly in the C1, domain.

The existence of this patch suggests that RAGE binds negatively charged fragments of the modified protein surface where AGE alteration either removed positive charges from the protein surface or added negative charges (e.g., in the case of CML or CEL) [69]. This hypothesis was initially confirmed by mutagenesis and NMR experiments that identified the involvement of Lys37, Lys43, Lys44, Arg48, Arg98, and Arg104 lining the positively charged patch in interactions with AGE-modified bovine serum albumin (BSA) [69]. Later, it was corroborated by the NMR structure of the V domain with a CEL peptide (Figure 3e) [74].

This large, positively charged patch is naturally a potential site for DNA binding. However, when analyzing the structure of RAGE with an oligonucleotide (PDB ID 4OI7) [71], the interface between the protein and DNA only partially overlapped with the positively charged groove. The DNA molecule formed polar contacts between the phosphate groups in the nucleic acid backbone and Lys or Arg side chains, precisely to lysines 107, 43, 37, 39, and 123 and to arginines 218, 29, and 216, most of which built the positive patch (lysines 107, 43, 37, and 39 and arginines 216 and 218). The remaining polar amino acid residues lining the pocket, i.e., arginines 114, 116, 104, 77, 98, and 48 and lysines 44, 110, and 52, did not form any contacts with the nucleic acid. As part of the positively charged patch remained unoccupied, perhaps it would be possible to simultaneously bind a negatively charged AGE and a DNA molecule, as a CEL moiety contacted the RAGE surface in the proximity of Lys110 and Arg98 (Figure 3e) [74].

Concerning RAGE’s biological roles, besides activating NF-κB and mediating inflammation, it regulates the expression of the cytokine TNF-α, oxidative stress [75], and dysfunction of the endothelium in type 2 diabetes [67]. As a multiligand receptor, it interacts with the amyloid-β peptide precursor (ABPP), mononuclear phagocytes and lymphocytes, S100/calgranulins (i.e. S100A12, S100B), and amphoterin [76]. This signalling cascade triggers the activation of the ERK1/2 and p53 pathways and mitogen-activated protein kinases (MAPKs). For example, interactions with the ABPP and S100B after myocardial infarction entangles this receptor also in atherosclerosis, cancer, and neurodegenerative diseases [77,78]. 

The soluble isoform of the receptor is a prospective biomarker of adverse outcomes and disease risk due to the fact that elevated sRAGE blood levels may reflect the amplification of pathological pro-inflammatory processes [79]. Moreover, the administration of this isoform accelerated wound healing in diabetic mice [76]. Finally, RAGE is one of the receptors associated with infection with SARS-CoV-2 and the progress of the COVID-19 disease [80,81,82].

### 2.2. Stab1 and Stab2

In the 1990s, Smedsrød and others published several papers concerning the scavenger function of liver endothelial cells. This included the elimination of AGEs by scavenger-receptor-mediated endocytosis in hepatic sinusoidal Kupffer and endothelial cells [83]. This research was further developed by the Smedsrød group in a study demonstrating that AGEs impaired the scavenger function of rat liver sinusoidal endothelia without naming the responsible receptors [84]. A year later, Tamura et al. demonstrated that Stab1 and Stab2 (or FEEL-1 and FEEL-2, fasciclin, EGF-like, laminin-type EGF-like, and link domain-containing scavenger receptor-1 and -2—the alternative names for Stab1 and -2) are the endocytic receptors for AGEs [55]. 

Human Stab1 and Stab2 are single-pass type I transmembrane glycoproteins belonging to class H scavenger receptors [57]. Full-length Stab1 and Stab2 have a molecular mass of 315 kDa, consist of 2551 and 2570 amino acids, respectively, and are coded by the STAB1 and STAB2 genes, respectively. Two active Stab2 isoforms of different molecular masses exist: the lighter isoform, 190 kDa, is formed by the proteolytic cleavage of the full length 315 kDa protein. Stab2 is highly expressed in the sinusoidal endothelial cells (SECs) of the liver, spleen, lymph nodes, bone marrow, and pancreas, where it plays an important role in homeostasis and immunity [85]. It is also found in non-SEC cells, such as human monocyte-derived macrophages (HMDMs) and myoblasts. Tissue-specific splice variants are expressed in the bone marrow, lymph nodes, spleen, and likely in other tissues [86]. High levels of Stab1 are found in the spleen, lymph nodes, liver, and placenta; it is also expressed in endothelial cells [87].

Both Stab1 and Stab2 act as scavenging receptors for acetylated low-density lipoprotein and bind Gram-positive and Gram-negative bacteria, playing a role in defense against bacterial infection. When Stab1 is inhibited in endothelial tube formation assays, there is a significant decrease in cell–cell interactions, suggesting its role in angiogenesis [87]. Stab1 is also involved in the delivery of newly synthesized CHID1/SI-CLP (stabilin-1-interacting chitinase-like protein) from the biosynthetic compartment to the endosomal and lysosomal systems [88].

Stab2 (with the alternative name of hyaluronic acid receptor for endocytosis, HARE) recognizes and mediates the endocytosis (via the clathrin/AP-2-dependent pathway) of multiple ligands, including hyaluronic acid (HA), heparin (Hep), chondroitin and dermatan sulfates (CS, DS), AGEs, acetylated low-density lipoprotein (AcLDL), and pro-collagen propeptides [57,89]. In general, the removal of HA and other glycosaminoglycans (GAGs) from the bloodstream is crucial for maintaining blood at an acceptable viscosity, and this is most efficiently performed by liver SECs.

Most ligand binding sites are located within the Link domain, a defining feature of Stab2 [57,89,90]. This domain of ca. 100 amino acids is found also in homologous HA-binding proteins engaged in extracellular matrix creation, cell adhesion, and migration, such as CD44, RHAMM, or TSG-6 [91]. Both isoforms of Stab2 bind HA with similar strengths, i.e., K_d_ = ~10–20 nM [89]. Stab1 also possesses the Link domain; however, it is not able to bind to a wide variety of ligands, including HA.

Apart from the Link domain, Stab1 and Stab2 contain 7 fasciclin 1 (FAS1) domains, 23 epidermal growth factor-like (EGF-like) domains in 4 series of repeats, laminin-type EGF-like domains, a transmembrane region, and a short C-terminal domain (Figure 4a) [57]. FAS1 domains are composed of ~140 amino acids and are present in many other membrane-anchored and secreted proteins, e.g., in periostin and βig-h3. In Stab2, FAS1 domains facilitate homophilic interactions leading to cell aggregation, presumably due to FAS1 dimerization [92] and heterophilic interactions mediated by binding to integrins, e.g., αMβ2 (the process of lymphocyte adhesion to the hepatic sinusoidal endothelium) [93]. An analysis of the interface between chains A and B in an asymmetric unit of the X-ray structure of the seventh FAS1 domain (Figure 3b; PDB ID 5N86) also revealed a potential mode of dimerization of FAS1 domains. However, the mechanisms of cell aggregation require further elucidation [94]. The cysteine-rich EGF-like domains were shown to directly and specifically recognize phosphatidylserine (PS) presented on the surface of apoptotic cells, primary necrotic cells, and erythrocytes [95]. Motifs enhancing the internalization of ligand–receptor complexes and their delivery to lysosomes were present in the short cytoplasmic C-terminus of Stab2 and facilitated the recruitment of adaptor proteins (GULP and thymosin-β4) involved in Stab2-mediated engulfment [96].

Even though the recognition of many diverse ligands of Stab2 has already been confirmed in several experiments, the data are still incomplete and gathered using different methods such as ELISA, surface plasmon resonance (SPR), fluorescence polarization (FP), or cell-based assays. Therefore, the ligand binding profile, especially in the context of binding sites and competition, still requires elucidation for this enigmatic receptor involved in the processes of muscle differentiation and regeneration [97], cancer [98], and diabetes [99]. What is especially thrilling among the recent findings about Stab2 is its emerging role in drug delivery into target cells. Thanks to the outcomes of the newest studies, the array of ligands for this receptor has turned out to be much wider and involved than it seemed before.

Stab2 is also present in the processes of the trafficking of drug carriers (HA-coated nanoparticles, carbon nanotubes) [100] and oligonucleotides [101]. Finally, the AGE binding site to Stab2 and the competition for internalization with other ligands remains unknown. This raises the question whether elevated circulating AGE levels could interfere with the potential application of Stab2 for trafficking therapeutic cargo into the cell. The only structural information available for the receptor is the X-ray structure of the seventh FAS1 domain (Figure 3b,c; PDB ID 5N86) [94]. None of the experimental structures of Stab1 are available, which obstructs homology modeling. Several X-ray or NMR structures of the Link domains of HA-binding proteins such as TSG-6 or CD44 with HA octamer are available (PDB ID 2JCR) [102]. However, the percent identities of the stabilin Link domain to its human homologues are not very high (e.g., 47% to the TSG-6 Link domain, PDB ID 2PF5). Despite some similarities in terms of the negative charge of certain AGEs to hyaluronic acid, it cannot be concluded that these ligands share the same binding site, as, for example, negative Hep and HA molecules do not [89]. The presence of a positively charged groove similar to that on the surface of RAGE on the surface of Stab2 could be a potential binding site for AGEs, but it could also be a pocket for HA, Hep, or other GAGs. The final verification can only be achieved by providing experimental structures of Stab2 and Stab1.

### 2.3. Other Scavenger Receptors

Other classes of scavenger receptors (namely A, B, and E) were also reported to bind AGEs [103].

Scavenger receptors from classes A, SR-AI, and transcript variant SR-AII mediate the endocytosis of a broad group of macromolecules, including modified low-density lipoproteins (LDLs) [58]. They are single-pass type II membrane glycoproteins encoded by the MSR1 gene implicated in the pathologic deposition of cholesterol in the arterial walls during atherogenesis. These receptors may mediate the detoxification of AGEs, but they have not been demonstrated to induce further signal transduction [76,104,105].

CD36 and SR-BI, multi-pass membrane scavenger receptors from class B, span the cell membrane twice and, with their extracellular domains, bind a diversity of ligands [67]. SR-BI binds high-density lipoprotein (HDL), mediates the selective uptake of cholesterol ester without subsequent endocytic uptake of apolipoproteins [106], and also recognizes PS and, consequently, apoptotic cells exposing PS [107]. It is also involved in the processes of microbial and viral infections, e.g., with *Mycobacterium fortuitum*, *Escherichia coli*, Staphylococcus *aureus* [108], hepatitis C virus [109], and human coronavirus SARS-CoV-2 [110]. CD36, in turn, recognizes anionic phospholipids, fatty acids, collagen, and oxidatively modified low-density lipoprotein (oxLDL) [111]. It is not directly responsible for the endocytic uptake of circulating AGEs, but it has a crucial function in the induction of cellular oxidative stress. This phenomenon was investigated in liver endothelial cells with glycolaldehyde-modified BSA (GA-BSA), methylglyoxal-modified BSA (MGO-BSA) [112], and 3T3-L1 adipocytes. The interaction of AGEs or oxLDL with these cells via CD36 induced oxidative stress as an intracellular signal and led to the abrogation of leptin expression [113]. It has been also demonstrated that AGE-modified proteins interfere with the SR-BI uptake of HDL, suggesting a possible pathological role of AGEs in the HDL-facilitated reverse cholesterol transport system [67,114].

Finally, a single-pass type II scavenger receptor from class E, lectin-like oxidized low-density lipoprotein receptor 1 (LOX-1), encoded by the OLR1 gene was found to bind oxLDL, HSP70 chaperone, and AGEs [60,115]. OxLDL is a marker of atherosclerosis, inducing cell activation and dysfunction of the vascular endothelium, which leads to pro-inflammatory responses, pro-oxidative conditions, and apoptosis. LOX-1, upon the binding of this ligand, triggers cellular events such as the activation of NF-κB through an increased production of intracellular reactive oxygen or a reduction in the concentration of nitric oxide in the cell [67,116]. The binding of AGE-modified proteins was demonstrated with bovine aortic endothelial cells and CHO cells expressing bovine LOX-1 in an experiment confirming inhibition by the anti-LOX-1 antibody [115]. AGE ligands have increased the expression of LOX-1 in the vascular endothelium of diabetic rats [103,104,117].

No experimental models of the structures of class A and B scavenger receptors are available. However, several X-ray structures for class E have been reported, out of which most present models without ligands (e.g., PDB IDs: first structure by Park et al. [118], 1YPO, followed by 1YXJ, 6TL7). The most relevant in the context of oxLDL recognition is a W150A LOX-1 mutant showing impaired binding (PDB ID 3VLG) [119].

### 2.4. AGE-R1, -R2, and -R3

Besides RAGE and several scavenger receptors, three other proteins are also able to recognize and bind AGEs followed by no signal transduction, possibly causing their clearance and detoxification: AGE-R1 (OST-48), AGE-R2 (80K-H), and AGE-R3 (Gal-3) [76,104,105].

AGE-R1 is an essential 48 kDa subunit of the N-oligosaccharyl transferase (OST) complex embedded in the endoplasmic reticulum. OST catalyzes the transfer of a high-mannose oligosaccharide (Glc_3_Man_9_GlcNAc_2_ in eukaryotes) from a lipid-linked oligosaccharide donor to Asn residue in the Asn-X-Ser/Thr consensus motif of the acceptor polypeptide chain. It is encoded by the DDOST gene and required for efficient N-glycosylation [120].

AGE-R2 is a regulatory subunit of glucosidase 2 anchored in the endoplasmic reticulum that sequentially cleaves the two innermost α-1,3-linked glucoses from the Glc_2_Man_9_GlcNAc_2_ oligosaccharide precursor of immature glycoproteins [121]. It is encoded by the PRKCSH gene and involved in N-glycan metabolism.

AGE-R3 is a galactose-specific lectin (galectin-3) that binds IgE, is encoded by the LGALS3 gene, and can be found in the cytoplasm or secreted. It is required with DMBT1 for the terminal differentiation of columnar epithelial cells during early embryogenesis. Galectin-3 also mediates stimulation by CSPG4 of endothelial cell migration with the α3β1 integrin [122].

Several individual structures of AGE-R1 (cryoEM) and AGE-R3 (X-ray) have been reported, but not in the context of AGE binding.

The expression of AGE-R1 was decreased in diabetic kidney disease [123], but its upregulation increased AGE removal [103]. The overexpression of AGE-R1 in transgenic mice protected them against diabetic nephropathy [124] by preventing AGE accumulation [125]. AGE-R1’s relationship with renal disease was also demonstrated in a study identifying it as a negative regulator of inflammatory responses to AGE in mesangial cells [126]. Nephropathy, in turn, is one of the diabetic complications linking AGE metabolism and AGE-R1 clearance with type-1 diabetes: a study demonstrated the harmful effects of RAGE modulation and the protection afforded by AGE-R1 in the context of diabetes [127].

Concerning AGE-R1 discovery, it was initially discovered that a p60 kDa protein was involved in AGE binding and removal. Later, it was identified that the plasma membrane component of p60 had a 95% similarity to that of OST-48, a subunit of the OST complex. This OST-48 N-terminal component of AGE-R1 was postulated to interact with AGEs [123,128].

The second receptor, AGE-R2, initially identified as an 80–90 kD component and found homologous to the protein kinase substrate 80K-H [128], is phosphorylated upon exposure to AGEs. Hence, it has been postulated that it contributes to raising AGE signaling in the early stage; thus, it was termed AGE-R2 [20,124]. This signaling induced cell activation and cytokine and growth factor secretion [105]. Thus, the interaction of AGEs with AGE-R2 led to adverse effects on cells, such as inflammation and altered metabolic effects [103]. Both AGE-R1 and AGE-R2 were detected in human atherosclerotic plaques with a distribution pattern similar to AGE accumulation in cells [105].

The binding of AGEs to AGE-R3 (known also as Galectin-3) increases the receptor’s expression, inducing a positive feedback loop by further increasing AGE–ligand binding and endocytosis by macrophages [20]. AGE binding to this receptor is characterized by a high affinity (K_d_ 3.5 × 10 7 M^−1^) and occurs within the 18 kD C-terminal fragment [129]. It promotes the formation of a heavy complex with the AGE ligand and other membrane-associated receptors on the cell surface, presumably via the attack on a thiol ester of AGE-R3 by AGE protein nucleophilic groups. These high-energy thiol ester bonds may be beneficial for the further trafficking of diverse, bulky AGEs into the cell for degradation [105].

Hence, AGE-R1 and AGR-R3 participate in the detoxification of AGEs. After endocytosis and the intracellular degradation of HMW-AGEs to LMW-AGE peptides, the AGE peptides re-enter circulation and undergo filtration in glomeruli to be further reabsorbed or excreted in the urine. The effective removal of AGEs is dependent on normal renal function, suggesting the association of renal disease with reduced excretion of AGEs [103]. AGE-R3-deficient diabetic mice suffered from glomerulopathy, proteinuria, and mesangial expansion, whereas the increased glomerular expression of this receptor was noticed in diabetic mice [124]. AGE-R3’s pathophysiological function was also noted in an aging heart. Age-dependent upregulation was detected for the gene coding of AGE-R3, with a subsequent downregulation of RAGE and no significant differences for the AGE-R1 and AGE-R2 genes [130].

## 3. Pathophysiological States and Diseases Induced by AGEs

The exact mechanism of carbohydrate-rich food action on human health is still unknown, and many studies focus on this issue. Carbohydrates from food change the levels of internal glucose, impair glucose pathways in tissues, and, in consequence, can strongly and negatively affect human health [53]. The presence of AGEs, also called ‘glycotoxins’, is the result of the accumulation of a high amount of sugar compounds and their interactions with human amino acids, peptides, and proteins. It represents one of the theories explaining the causes of cell damage and several tissue disorders, such as stiffening of the extracellular matrix due to collagen glycation or agglomeration of glycated β-amyloid leading to AD [131].

AGEs often link with long-lived proteins, such as collagen, further extending the life of the protein. Due to the covalent linking of Lys and Arg residues with sugars and the acquirement of agglomerating properties, AGEs become physical barriers for tryptic cleavage and inhibit the degradation of the peptides [132]. Even despite the fact that AGEs do not glycate Phe, Trp, and Tyr residues participating in pepsin cleavage, the level of digestion with this enzyme is reduced, which was proved during gastric digestion of β-casein and β-lactoglobulin [133]. In addition, crosslinking and agglomerating can change the functions of a trapped protein [134].

The presence of AGEs in blood is a risk factor for diabetes mellitus, the most common disease connected with circulating carbohydrates [127,135]. Other diseases and disorders, such as retinopathy, neuropathy, kidney diseases, CVD and cerebrovascular diseases (CeVD), dermatology problems, and viral infections, are considered to be a direct or indirect effects of AGEs. Furthermore, there is evidence on the existence of a feedback loop where high amounts of glycotoxins cause the diseases, but most of these diseases hinder the excretion of AGEs from organisms and increase their endogenous production [136].

### 3.1. Absorption

The digestion of dAGEs starts by oral intake; then, most of these compounds are directed to the gastric tract and intestinal phases where absorbable AGEs are delivered to the circulatory system and their remains are excreted in urine. Non-absorbable compounds go to the lower gut where they are partially digested by gut microbiota and the remains are excreted with feces [136]. Protein- and peptide-bound AGEs are hydrolyzed by pepsin and pancreatin before the intestinal phases in the gastric tract. Although intestinal digestion by trypsin is inhibited due to crosslinking, it was proved that the success of proteolysis depends mostly on the protein structure, and sometimes glycation can even improve hydrolysis, which was proved with lactoferrin [137,138]. The level of absorption is different for various AGE structures and is particularly dependent on molecular weight. The absorption of AGEs depends also on hydrophobicity. AGEs with higher hydrophobicity (e.g., pyrraline and argpyrimidine) cross the basolateral membrane much easier than more hydrophilic AGEs (e.g., CML, CEL, and MG-H1) [48].

Zhao et al. proposed four hypothetical routes of AGE absorption: peptidase transporter PEPT1, transcytosis, paracellular, and simple diffusion [137]. Free AGEs penetrate the circulatory system via simple diffusion, which is not very effective; therefore, a part of them remains in the gastrointestinal tract. AGEs such as CML and CEL are absorbed into intestinal cells in the form of dipeptides, most likely by peptidase transporter PEPT1. The dipeptides are hydrolyzed into amino acids and go further to the basolateral membrane [48]. Studies have suggested that proteins bound to CML are hydrolyzed during intestinal luminal digestion to absorbable peptides [139,140]. Protein-bound AGEs have a higher affinity to RAGE than peptide- and amino-acid-bound AGEs, which makes them more difficult for elution from the bloodstream, digestion, and purification by the kidneys [134].

#### 3.1.1. Absorption—Gut Microbiota 

Approximately 80% of dietary pre-AGEs, Amadori rearrangement products, cannot be absorbed by human tissues [141]. There is some evidence that part of the non-absorbed AGEs are digested by lower intestinal gut microbiota [142,143,144]. These microbiota have different enzymatic abilities than humans and degrade AGEs in different ways (e.g., via fermentation). Bacteria are able to use dAGEs remaining after human digestion for their own metabolism, although a study has suggested that LMW dAGEs are digested more easily than HMW ones [137]. However, the presence of dAGEs in the lower gut is not indifferent to the microorganisms living there [145,146]. Snelson et al. demonstrated that high levels of AGEs in food resulted in a decrease in the amount and variety of bacterial strains from the genera *Bacteroides*, *Bifidobacteria,* and *Lactobacilli* [146].

The two-stage degradation of AGEs by gut microbiota begins with the consumption of sugars and continues with the decarboxylation of amino acids to biogenic amines, which are further degraded by microorganisms or released to the human lower gut. This metabolic pathway is common in many bacterial genera belonging to *Bifidobacterium*, *Clostridium*, *Enterococcus*, *Lactobacillus*, *Pediococcus*, *Streptococcus*, *Enterobacter*, *Escherichia*, *Klebsiella*, *Morganella,* and *Proteus*. The production of biogenic amines from amino acids is one type of cellular protection from the acidic environment of the large intestine, where the pH value is 5.5–6.8. The structures of the amines are dependent on strains and amino acids [147]. The other pathway of degrading amino acids from AGEs begins with transamination to α-keto acids and then their oxidative decarboxylation to aldehydes. The next step is reduction to alcohols or oxidation to fatty acid [142].

A study showed that intestinal bacteria from healthy adults are able to degrade CML in anaerobic conditions in two ways: partly (by *Clostridium butyricum*, *Clostridium sphenoides*, *Escherichia coli*, or *Oscillibacter* spp.) or entirely (*Cloabacillus evryensis*) [143]. Hellwig et al. reported that *E. coli* was able to degrade CML to different products, mainly to the biogenic amine N-carboxymethylcadaverine (CM-CAD), and this metabolism depended on the amount of oxygen in the gut environment [142].

#### 3.1.2. Excretion

In healthy people, about one-third of AGEs are excreted in urine, while in individuals with diseases such as diabetes mellitus, it is less than 5% [148]. It is known that approximately 15–25% of CML is excreted in urine, proportional to oral intake. However, the kidney capability of CML elimination is limited, and after exceeding this capability, the higher the level of consumed CML, the lower the ratio of CML excreted renally. This suggests the constant rate of kidneys to filtrate AGEs. In contrast, the fecal excretion of CML is proportional to intake without limitations. A study from 2012 showed that the ratio of excretion to consumption was 31.7% in an AGE-rich diet and 22.5% in an AGE-low diet [149].

Studies on urinary excretion have also been carried out on dogs and cats. The more processed food the animals were fed, the bigger the ratio of AGEs detected in their urine. Furthermore, it was observed that, despite feeding dogs and cats with raw food, the levels of excretion of CML, CEL, and lysinoalanine (LAL) were still high, which suggests that diet is not the only source of AGEs and that some of them are produced endogenously [150]. Although the results of this experiment were complimentary with human studies, work on animal models cannot be directly extrapolated because of differences in their gastrointestinal tracts. Absorption has many molecular variables (such as protein activities and tissue functions) that make in silico research inconsistent, and an initiative has been taken to organize the protocols and guidelines of measure for common parameters. The international network INFOGEST brings together scientists from around the world who work on static in vitro models to allow for maximum standardization of the results obtained. They proposed the INFOGEST digestion protocol in 2014 [151]. Its success contributed to further improvements, and an updated version of the protocol was published in 2019 [152]. One of the recently proposed models is the TNO gastrointestinal in vitro model (TIM-1). It enables researchers to observe much of the human gastrointestinal tract and also uses several parameters that have been problematic to reconstruct, e.g., dynamic pH curves, peristalting mixing, the action of bile and pancreatic enzymes, and passive absorption [153].

#### 3.1.3. Mechanisms of Action

Two basal mechanisms that explain the role of AGEs in diseases and disorders are: (i) the covalent crosslinking of serum and extracellular matrix (ECM) proteins, lipids, and DNA, resulting in their biochemical impairment and cell disruption; and (ii) interaction of AGEs with their receptors, especially on the AGE–RAGE axis, launching a series of cascade reactions and signaling pathways, proliferation, autophagy, and apoptosis [154]. There are also four more modes distinguished by Chen and Guo: (i) ROS generation (responsible for oxidative stress); (ii) mitochondrial function impairment; (iii) AGEs as antigens inducing immune responses; and (iv) allergies caused by AGEs [141].

### 3.2. Oxidative Stress and Inflammation States

The increase in oxidative stress and inflammation states by AGEs has three main pathways, consistent with the mechanisms of action mentioned above: (i) AGEs bind to the cell surface or crosslink with extra- and intracellular proteins; (ii) AGEs catalyze ROS formation and accumulation; and (iii) AGEs bind to receptors—especially RAGEs—which induce the cascades of pathogenic mediators [134]. It was demonstrated that the AGE–RAGE axis activated NADPH oxidase in mouse cells, which increased oxidative stress and inhibited transcription factor activity [155]. Earlier, in 1994, a negative impact of ROS induced by AGEs on endothelial cells via generating thiobarbituric acid reactive substances and activation of the NF-κB pathway was confirmed [156]. It was also demonstrated on a mouse murine mesengial cell line that the GOLD–RAGE axis induced mitochondrial function disruption via ROS generation, and it was considered one of the reasons for kidney failure and renal diseases during diabetes mellitus. Correlation between oxidative stress and inflammatory states was confirmed, while the inflammation state decreased when the antioxidant N-acetylcysteine (NAC) was added [157].

#### 3.2.1. Oxidative Stress

Oxidative stress is related to the amount of ROS in an organism. There are three mainly distinguished types of ROS: superoxide anion radical (O_2_^−^), hydrogen peroxide (H_2_O_2_), and hydroxyl radical (HO·). There are also oxidants containing nitrogen, and the most common in studies of AGEs in human organisms is nitric oxide (NO·). The group of nitrogen oxidants is called reactive nitrogen species (RNS). Their accumulation causes several problems, such as redox signaling and control impairment or molecular damage. Oxidants cause oxidative stress, which induces lipid peroxidation and glycoxidation reactions via attacking free amino groups in proteins, resulting in the formation of ALEs and AGEs, respectively [158].

A comprehensive study conducted on human keratinocyte cells incubated with nine different AGEs showed that structures based on 1,2-dicarbonyls (3-DG, 3-DGal, 3,4-DGE, GO, and MGO) induced cytotoxicity and oxidative stress (measured by peroxide, peroxynitrite, and superoxide production). Moreover, different types of AGEs induced different types of programmed cell death. MGO caused caspase-mediated apoptosis, while 3-DGal and 3,4-DGE induced the pathway based on the NF-κB factor and caused pyroptosis [159].

#### 3.2.2. Inflammation States

The proinflammatory potential of AGEs is connected with aging. Biochemical changes of cells and tissues associated with aging are often proinflammatory factors, and this is the reason for defining AGEs with the new term of ‘inflammaging’. As a result of this connection, many biomarkers of aging have been used in studies of inflammation states; levels of C-reactive protein (CRP), interleukin-6, ferritin, and lymphopenia have been mentioned [135,141]. The long-term consumption of AGEs results in the production of the pro-inflammatory effector molecule complement 5a; increases other inflammatory markers, such as IL6, IL-1a, ICAM, and TIMP-1; and leads to immune activation, which was confirmed with rodents [160]. Due to the fact that the blocking of RAGE by AGEs causes the impairment of innate immunity, a hypothesis of targeting AGEs in activated macrophages as therapy for AGE-related diseases was proposed [161]. There were some suggestions that the possible mechanism responsible for diseases derived from inflammation states is AGEs glycating long-living ECM proteins, producing a long term impact on the immunity system [162].

Several studies on human nucleus pulposus cells, fibroblasts, and pancreatic cells have reported that an inflammation state can be a result of activation of the NLRP3 inflammasome via ROS production or damage of pancreatic islets [163,164,165]. On the other hand, Son et al. reported that the damage of the NLRP3 inflammasome on macrophages caused the inflammation state [166].

### 3.3. Measurement of AGEs

Despite many studies on AGEs, their absorption, and their negative effect on human health, there are no systemized analytical methods to determine the amounts of AGEs in the blood, cells, and tissues. Studies have been carried out in many fields, and many variables have been analyzed, but most of them cannot be directly compared with each other [154]. One of the difficulties of measuring changes in endogenous AGEs in the human body is the naturally high background levels. Hellwig et al. suggested that the most reliable quantitative method of measuring AGEs, especially CML, is high pressure liquid chromatography with tandem mass spectrometry (HPLC-MS/MS). At the same time, they recognized ELISA as uncertain, even when repeated [142].

During studies on AGEs in the gastrointestinal tract, many factors must be simplified while performing an in vitro assay. This entails the necessity to ‘stabilize’ the conditions when, in fact, in the organisms they are very dynamic. For example, after the intake of food, the pH of the human stomach is approximately 1–1.5., During digestion, pH increases to 5–7. After about 3 h, pH returns to the initial range of 1–1.5. In many studies, the pH is averaged to 2–2.5 for approximately 2 h. Due to this, the TNO gastrointestinal in vitro model (TIM-1) of gastric digestion has been developed [153,167].

Skin autofluorescence (SAF) allows the detection of the amount of AGEs in patients with peripheral artery disease (PAD) [168], diabetic foot ulcers (DFUs) [169], or carotid atherosclerosis [170]. SAF is a method of measuring AGEs with the use of the fluorescent features of these compounds. The advantages of this method over measuring the amount of AGEs in the circulatory system or via skin biopsy are the information given about tissue AGE accumulation, the non-invasive method, and low costs. Most SAF measurement is conducted via AGE Reader (DiagnOptics). The initial versions of the software did not take skin color into account during measurements, so its improvements were focused on this issue and were soon introduced. The score is expressed as the ratio of light intensity in the 420–600 nm wavelength range to light intensity in the 300–420 nm wavelength range [171,172].

### 3.4. Diseases

#### 3.4.1. Diabetes Mellitus

Diabetes mellitus can develop into a number of complications, such as retinopathy [173], nephropathy, bone strength weakness [174], neuropathy [175], atherosclerosis [176], and CVD [177]. The glycation of HSA indicates a hyperglycemic state. AGEs are a possible factor for causing diabetes states and are believed to be a good biomarker of this disease. Glycotoxins are able to influence drug carriers, creating a necessity for extensive studies on diabetes, its etiology, and its related pathways [13].

Chronic oxidative stress and an overload of AGEs in serum, despite acting on many molecular pathways [178], can inhibit two receptors, AGE-R1 and SIRT-1, which are responsible for recognizing and repressing AGEs [179]. On the 3T3-L1 cell line, it was demonstrated that AGE-R1 and SIRT1 coregulated adipocyte insulin signaling during MGO treatment. On the other hand, a study on mice confirmed that MGO intake decreased the amount of serum mRNA of AGE-R1 and SIRT1, which led to decreasing insulin efficiency [180].

The presence of AGEs during gestational diabetes mellitus was confirmed. It is known that AGEs in gestational diabetes mellitus increase the concentrations of TNF-α and hs-CRP, which are responsible for inflammatory reactions [181].

Studies to determine the most accurate and reliable AGE biomarkers for diabetes represent a growing need. They would help in the diagnosis of hyperglycemic states and, consequently, early initiation of treatment and prevention. Chiu et al. performed studies testing twelve different biomarkers (ten AGEs and two oxidative markers), but only one passed the statistical reliability study: glucosepane (GSP). This compound was significant to type 1 diabetes mellitus patients. MG-H1 was suggested as a potential biomarker for diabetes, but more studies are required to confirm this candidate [182]. Heidari et al. suggested that a quantitative measurement of AGEs in the blood could be a biomarker of the duration of diabetes. In the group with long-lasting disease, the level of glycotoxins was the highest, while newly diagnosed patients had the lowest AGE concentrations [11].

#### 3.4.2. CVD and CeVD

CVDs, such as coronary heart disease (CHD), CeVD, and chronic heart failure (CHF), are often listed as complications during diabetes mellitus. The most important mechanisms of AGE action are: (i) the crosslinking of elastin and collagen, which leads to myocardial stiffening of the blood vessels and cardiac fibrosis [183]; (ii) the interaction between AGEs and RAGE resulting in NADPH oxidase activation, ROS release [184], and inducing the NF-κB pathway; (iii) the inhibition of nitric oxide activity and the reduction of endothelial nitric oxide synthase activity [185]; and (iv) an increase in vascular permeability [41,186,187].

A decrease in AGEs causes lowering in the expression of RAGE, the number of inflammatory cells, intercellular adhesion molecule-1 (ICAM-1), and vascular cell adhesion molecule-1 (VCAM-1; the factor of CVD) [188]. An elevated amount of 3DG-H1 with decreased amounts of methionine sulfoxide increases the risk of CVD. Higher CML and G-H1 levels in serum are characteristic for CVD pre-existing with diabetes mellitus [189]. Botros et al. tested the correlation between lifestyle factors, such as drinking coffee and smoking cigarettes, HDL and LDL levels, and the amount of AGEs measured by SAF (AGE Reader). No direct impact of cholesterol on the amount of AGEs was found, although cholesterol itself is known to contribute to CVD [36]. Because the AGE–RAGE axis is one of the mechanisms of glycotoxin action in organisms, the sRAGE serum level is likely to become a biomarker for CVD. It was found that the level of plasma sRAGE was significantly lower in nondiabetic postmenopausal Indian women with coronary artery disease (CAD) compared with the healthy control [190]. Similar results were obtained with nondiabetic patients with premature CAD and nondiabetic CAD men compared with healthy subjects [191,192].

It was found that, in patients with major β-thalassemia, the most severe form of CAD, the amounts of AGEs and other factors, such as the level of blood glucose, were increased. This suggested that AGEs can be involved in the pathogenesis of β-thalassemia, but there is a need for further studies [193]. RAGE was studied as a vasculopathy biomarker for sickle cell disease due to the fact that, in the plasma of patients, sRAGE levels were significantly increased in comparison with the control group [191].

#### 3.4.3. Neurodegenerative Diseases

The most common neurodegenerative diseases include AD, Parkinson’s disease (PD), amyotrophic lateral sclerosis, and prion diseases. They are characterized by a slow loss of neural tissue or neurons and the deposition of aggregated proteins in the brain. The deposition of several compounds, such as extracellular amyloid β-peptide plaques and intracellular neurofibrillary tangles (NFTs) built mainly from τ-protein, α-synuclein, and the pathological form of cellular prion protein, is specific for the AD, PD, and prion diseases, respectively [192,194]. The presence of AGEs leads to mitochondrial dysfunction and cognitive impairment by aggregating adducts and creating oxidative stress [195]. The brain is characterized by high oxygen consumption due to the functions performed and high glucose consumption; therefore, oxidative stress plays an important role in the development of neurodegenerative diseases [196]. Moreover, the presence of AGEs was confirmed in Lewy bodies, inclusion bodies in the nerve cells, the excess of which are considered to be one of the causes of PD. It is also a marker of this disease [197]. Estrogen is an antioxidant, and a study showed that an increased level of glycotoxins associated with neurodegenerative diseases is dependent on gender [198].

In AD, AGEs enhance the aggregation of glycated proteins, which increase crosslinks in various parts of the brain. This increases the level of β-amyloid aggregation in senile plaques. AGEs were also shown to contribute to amyloid precursor protein (APP) processing, enhancing the cell-death-related pathway and upregulating SIRT1 expression, which impairs the protective functions of SIRT1 in neuronal cells [199]. Additionally, AGEs may lead to hyperphosphorylation of the τ-protein to form aggregates and further the formation of hyperphosphorylated microtubuli, NFTs, which are able to induce oxidative stress [200]. Moreover, related to presence of AGEs in serum, RAGE (along with the low-density lipoprotein receptor-related protein LRP) is one of the two transporters responsible for the passage of undegraded β-amyloid through the blood–brain barrier [201].

The aggregation of the α-synuclein contributing to the development of PD is facilitated due to the fact that the protein has many easily glycated Lys residues in its sequence. Through the formation of crosslinks, AGEs contribute to an increase in the amount of aggregated α-synuclein and, consequently, to the development of PD [202].

The increased amount of AGEs in neurodegenerative diseases suggests that they may be utilized as biomarkers of these diseases, but there is a need to improve their measurement methods. Also, sRAGE was studied in terms of a potential biomarker because its confirmed increased amount allowed the detection of the early risk stage [203].

#### 3.4.4. Kidney Diseases

Urinary excretion is one of the primary ways of eliminating glycotoxins from an organism. The renal proximal tubule cells that absorb AGEs from the glomerular filtrate are prone to the increased presence of these compounds [204]. Due to the harmful effects associated with their presence in the kidneys, AGEs belong to a group of uremic toxins. Injury can lead to chronic kidney disease (CKD) development and, consequently, to end-stage renal/kidney disease (ESR/KD) and death. Diseases comorbid with CKD include diabetes [205] and sarcopenia [206]. The negative impact of AGEs, in addition to causing inflammation states and oxidative stress via the AGE–RAGE axis, is related to vascular stiffness as a result of forming crosslinks in collagen structure [207] or the calcification [208,209] and metabolization of glycotoxins by human gut microbiota to other uremic toxins, such as trimethylamine N-oxide (TMAO) [210]. On the other hand, some probiotic microorganisms from human gut microbiota, such as the *Lactobacilli* family, are known to produce glyoxalase, an enzyme that degrades dAGEs [211].

One of the AGE receptors, AGE-R1, plays a protective role in kidney disease, but the nature of these ligand-and-receptor bindings is not fully understood. There was a suggestion that AGE-R1 is responsible for the clearance of glycotoxins from the circulation system and the mediation of AGE excretion by the kidneys [123].

In vivo studies in animals demonstrated that RAGE was the main factor responsible for the development of kidney disease. RAGE-knockout mice did not suffer from nephrosclerosis lesions and renal senile amyloidosis and were better protected against inflammation states and oxidative stress [212].

Potential biomarkers for progressive diabetic kidney disease include general AGE serum indices, the amount of CML [213], or hydroimidazolones [214].

#### 3.4.5. Collagen Tissues

Bunches of the characteristic collagen structures (triple helices composed of three α-chains) stabilized by the presence of Gly residues at every third position form in highly hydrated collagen fibrils. The accumulation and crosslinking of AGEs caused by oxidation stiffen the collagen structure and collagen-rich tissues such as tendons, bones [215], or the keratoconus. Damage to these tissues is the result of fractures of the inelastic collagen structure. A negative effect is also decreasing fibril slippage and viscoelastic properties [216]. Another theory about the reason of collagen fibril fragility is excessive hydration of the protein structure caused by AGE attachment [217].

There was a suggestion that, due to AGE crosslinking, accumulation and proliferation of misfolded collagen fibers can cause idiopathic frozen shoulder disorder [218]. Furthermore, such distortion in the structure of triple helices results in a reduction in the diameter of collagen fibrils and, as a consequence, in delayed bone healing disorder, which was observed in mice on an obesity-inducing diet [219]. A report indicated that the induction of oxidative stress and inflammation states by glycotoxins had an impact on skin collagen and could cause psoriasis [220]. On the other hand, damages to collagen-rich tissues can be caused by its apoptosis activated by the AGE–RAGE axis, leading to the induction of the p-p38 MAPK and NF-κB-p-p65 pathways [221]. The apoptosis of collagen fibroblasts slows wound healing, which was noticeable in patients with high levels of serum AGEs, mainly in diabetics and elderly people [222].

#### 3.4.6. Cancer

There are studies proving that, among many others, chronic inflammation state is directly connected with cancer development in different cells and tissues [223,224]. Inflammation, hypoxia, and oxidative stress are microenvironmental factors of excessive proliferation caused by ligand–RAGE interaction. These three states result in the activation of the signaling pathway of NF-κB [225]. Many circumstances related to the metastasis of cancer, such as diet, chronic diseases, and obesity, are also associated with the increased accumulation of AGEs in pathologically changed tissues [226].

Several mechanisms of AGE–RAGE-axis action leading to carcinogenesis were proposed by Malik et al.: (i) enhancement of the proliferation of cancer cells and activation of numerous pathways by AGE–RAGE, resulting in exacerbating the oncogenesis in the cells; (ii) activation of the survival response of cancer cells by inhibiting the apoptotic processes and spreading immortal cells around the tissues; and (iii) overexpression of RAGE in tumor cells and its interaction with ligands other than AGEs (S100 proteins). The activation of pro-survival and anti-apoptotic peptides (limiting the free transport of pro-apoptotic proteins such as p53 around the cell), the induction of autophagy, and angiogenesis are possible explanations for the second type of mechanism [227].

Several population studies have reported that an increasing level of AGEs in the blood, either through dietary intake or as a result of a disease such as diabetes, enhances the risk of advanced-stage breast cancer [228], prostate cancer [229], hepatocellular carcinoma [230], or oral squamous cell carcinoma [231]. Also, mice models of lung carcinogenesis in vivo exposed higher levels of serum CML in comparison with control groups [232]. The impact of CML on tumor promotion was also studied in mouse models of human pancreatic cancer (in vivo) and on the pancreatic ductal adenocarcinoma (PDA) cell line (in vitro). The presence of CML caused PDA cell proliferation and stimulated RAGE expression and, as a consequence, the AGE–RAGE axis [233]. Lee et al. studied the impact of AGEs on triple-negative breast cancer cells by the application of antibodies against several proteins. The team proved that AGEs are responsible for upregulation of the MMP-9 expression via activation of the ERK and MMP-9 promoters in the NF-κB pathway. An activation of the ERK and NF-κB pathways enhanced tumorigenicity, invasion, and transport of the cancer cells [234]. Two cell lines of colorectal cancer, with or without Glu-AGEs, were chosen for examination. The study showed that AGEs, by interacting with RAGE, enhance the expression of the general transcription factor and its target gene, Sp1 and MMP-2, respectively, which promotes metastasis, migration, and invasion of cancer cells [235].

#### 3.4.7. Liver Diseases

One of the roles of the liver is to catabolize and remove AGEs from an organism using liver SECs and Kupfer cells. This results in the agglomeration of glycotoxins when the organ suffers from any disorder. Moreover, the liver is sensitive to effects caused by the presence of AGEs in an organism, such as the inflammation state, oxidative stress, and the induction of the apoptosis signaling pathway. All of the above mentioned lead to cellular dysfunction, steatosis, and NAFLD [236].

Liver cells are sensitive, not only to common oxidative or nitrosative stress caused by the generation of harmful ROS and RNS, but also to carbonyl stress caused by reactive carbonyl species, particularly α-dicarbonyls, which include some AGEs (e.g., GO, MGO, and their derivatives). Their presence is associated with higher levels of RAGE in the liver for NAFLD and liver cancer progression [237].

The proposed biomarkers for monitoring NAFLD progression are plasmatic fluorescent AGEs measured and quantified via spectroscopy [238] and sRAGE, low concentrations of which may indicate injury of the liver. However, in the case of the latter, determining only this one parameter is nonspecific, so additional liver markers such as the Fibrosis-4 (FIB-4) index are necessary to assess the progression of liver diseases [239].

#### 3.4.8. Infertility

Infertility is listed as a complication of diabetes mellitus; however, its diabetes-independent relationship with AGEs has not been fully established yet. The causes of infertility in women and men differ in terms of body structure. In the former group, the main cause is ovarian dysfunction, and in the latter, testicular dysfunction and abnormal sperm function. Three diseases are listed among the components of female infertility: polycystic ovary syndrome (PCOS), endometriosis, and ovarian aging. The diagnostic criteria accompanying these ailments are hyperandrogenemia, rare ovulation, or long-term anovulation [240].

Increased amounts of AGEs cause oxidative stress in the ovarian cells and change the levels of ovarian steroid hormones, which are considered to be one of the causes of their negative impact on human fertility [241]. However, the effects of AGEs are not limited to enhancing oxidative stress because they also activate the AGE–RAGE axis pathways leading to inflammation states. Inflammation contributes to the damage of ovarian function, as demonstrated by testing mice fed with a high-AGE diet for twelve weeks [242]. The accumulation of glycotoxins in follicles enhances the inflammation state in human granulosa-lutein cells. Inflammation contributes to poor oocyte morphology and poor embryo development [243]. Male infertility may begin with oxidative stress caused by AGEs, as a study confirmed that the polyvalent unsaturated fatty-acid-rich membrane and DNA of sperm are very sensitive to the presence of ROS [244]. AGEs were not noticed to damage sperm cells or affect their motility [245].

The exact relationship of AGEs with androgens has not been established. However, it was noted that, by increasing the endoplasmic reticulum stress in granulosa cells, hyperandrogenemia caused an increased expression of RAGE and accumulation of glycotoxins [246].

Since both AGEs and sRAGE [247] were detected in the follicular fluid and it was shown that their presence influenced infertility, the effect of glycotoxins on embryos has aroused an interest. AGEs showed embryo-toxic properties in rat cells [248]. However, no significant statistical relationships have been found between the course of pregnancy to live-birth ratio with AGE levels in the follicular fluid [249].

#### 3.4.9. Viral Infection—COVID-19

During SARS-CoV-2 infection, the most important intrusion route into the cell is through the cellular transmembrane angiotensin-converting enzyme 2 (ACE2) molecule—receptors widely distributed in tissues. ACE2 expression is triggered by the development of inflammation, which may begin with an increased expression of RAGE caused by the presence of AGEs and their common AGE–RAGE axis [82,250]. Furthermore, AGEs affect a number of diseases that increase the risk of severe SARS-CoV-2 infection, such as inflammaging, diabetes, hyperglycemia, hypertension, heart disease, and obesity. The overexpression of RAGE also affects the lungs, a common target tissue attacked by SARS-CoV-2. Increased levels of this receptor and its mediator role in the lungs have been demonstrated in such respiratory diseases as allergic airway inflammation, asthma, pulmonary fibrosis, lung cancer, chronic obstructive pulmonary disease, acute lung injury, pneumonia, cystic fibrosis, and bronchopulmonary dysplasia [81,251].

Recent studies have suggested that the level of sRAGE should be considered as a biomarker in determining the need for high-flow nasal oxygen or mechanical ventilation in COVID-19 patients. Although the increased levels of various RAGE isoforms such as RAGE, esRAGE, cRAGE, and the cRAGE/esRAGE ratio are not COVID-19 specific, it seems that the general trend in the course of this disease is independent of comorbidities [252,253].

## 4. Conclusions

In the last decades, AGEs, constituting a heterogeneous, chemically diverse, and complex group of compounds formed either exogenously or endogenously, have gained the interest of the scientific community due to increasing evidence of their engagement in many pathophysiological states and diseases. As the complexity of their involvement is far beyond simple studies of the isolated signaling pathways, further research should aim to expand the knowledge on AGE formation, receptors, and competition with other ligands (e.g., in the case of multiligand receptors such as RAGE or Stab2). The structural basis of AGE interaction with receptors is limited, and providing the experimental structures of these complexes would undoubtedly facilitate the understanding of their still enigmatic roles. Another field of research is the ways of preventing elevated AGE levels, which cause pathologic states and diseases in the human body. Among the so-far developed concepts, the reduction of excessive dAGE intake by avoiding certain food processing (boiling or steaming rather than grilling or frying), avoidance of a high-sugar diet [31,49], or consumption of natural phytochemicals and phenolic compounds [6,254,255] are the leading ideas for the inhibition of adverse dAGE action. It is not surprising that these strategies are in accordance with the guidelines for a modern, healthy diet, and potentially the official directives could be refined more explicitly by stating the role of the excessive consumption of dAGEs for human health. In the light of growing evidence of AGE entanglement in many cellular processes, this complex topic will certainly remain intriguing for the research community.

## Figures and Tables

**Figure 1 cells-11-01312-f001:**
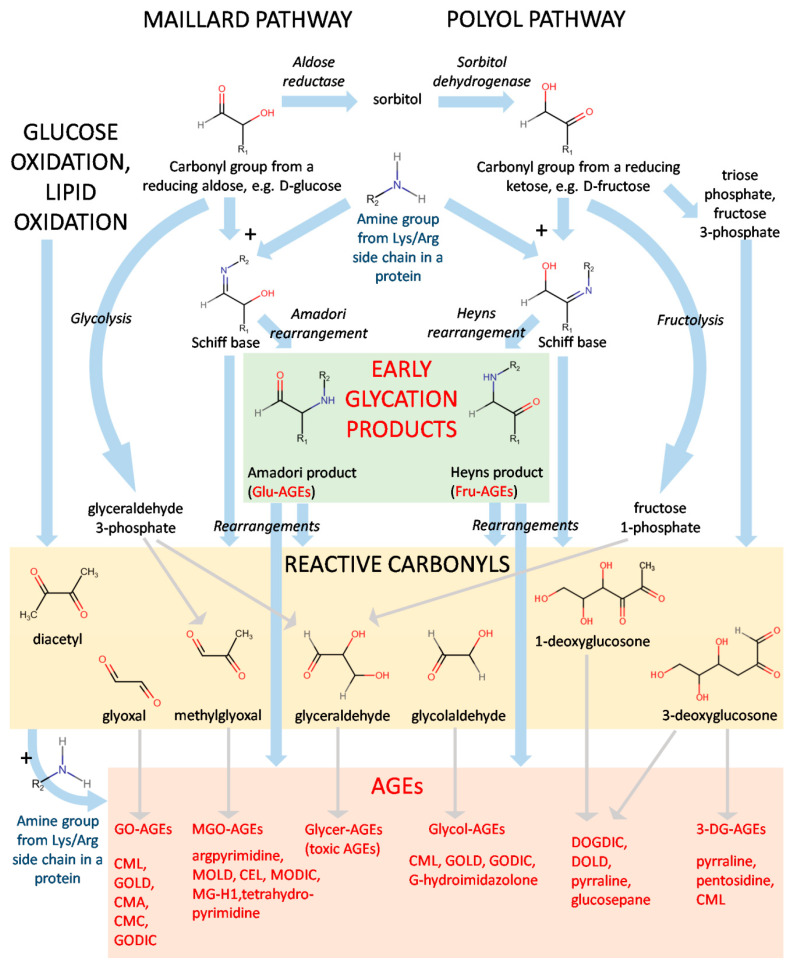
A scheme of formation for AGEs. Abbreviations: glucose-derived (Glu-AGEs), fructose-derived (Fru-AGEs), glyoxal-derived (GO-AGEs), methylglyoxal-derived (MGO-AGEs), glyceraldehyde-derived (Glycer-AGEs), glycolaldehyde-derived (Glycol-AGEs), and 3-deoxyglucosone-derived (3-DG-AGEs).

**Figure 2 cells-11-01312-f002:**
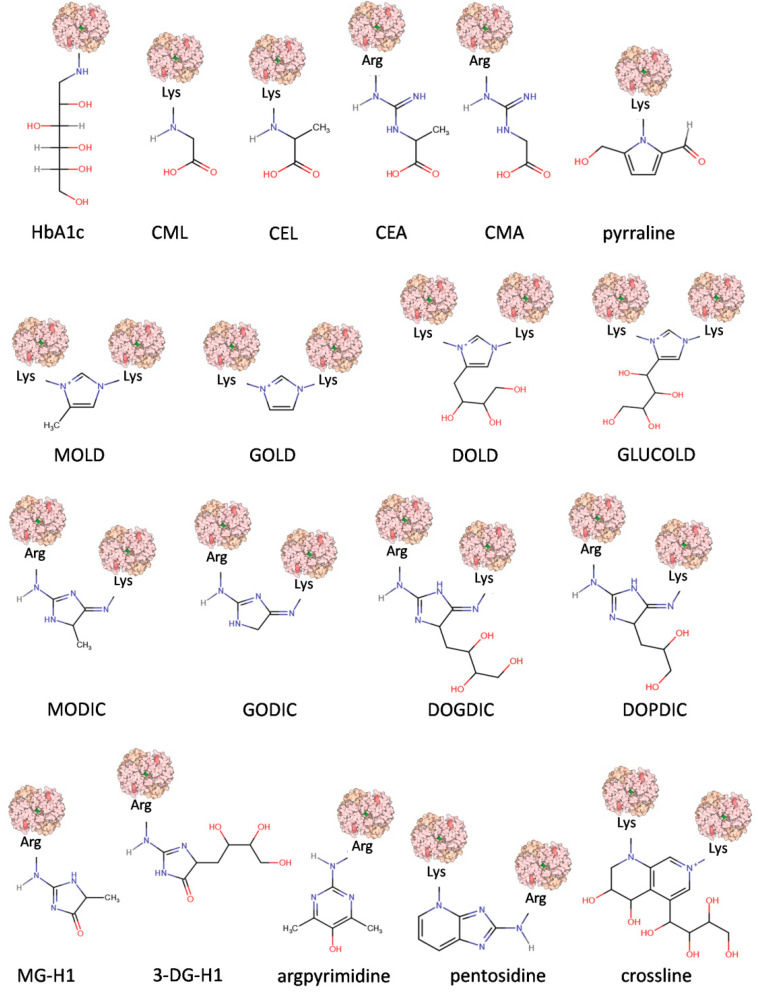
Chemical structure representations of AGEs. Abbreviations: glycated hemoglobin (HbA1c), N^ε^–(carboxymethyl)lysine (CML), N^ε^-(1-carboxyethyl)lysine (CEL), N^7^–(1-carboxyethyl)arginine (CEA), N^7^–(carboxymethyl)arginine (CMA), 6-(2-formyl-5-hydroxymethyl-1-pyrrolyl)-L-norleucine (pyrraline), 6-{1-[(5S)-5-ammonio-6-oxido-6-oxohexyl]-4-methyl-imidazolium-3-yl}-L-norleucine (MOLD), 6-{1-[(5S)-5-ammonio-6-oxido-6-oxohexyl]imidazolium-3-yl}-L-norleucine (GOLD),1,3-di(N^ε^-lysino)-4-(2,3,4-trihydroxybutyl)-imidazolium (DOLD),1,3-bis-(5-amino-5-carboxypentyl)-4-(1,2,3,4-tetrahydroxybutyl)-3H-imidazolium (GLUCOLD), 2-ammonio-6-({2-[4-ammonio-5-oxido-5-oxopently)amino]4-methyl-4,5-dihydro-1H-imidazol-5-ylidene}amino)hexanoate (MODIC), N^6^-(2-{4S(-4-ammonio-5-oxido-5-oxopentyl]amino}-3,5-dihydro-4H-imidazol-4-ylidene)-L-lysine (GODIC), N^6^-{2-{[(4S)-4-ammonio-5-oxido-5-oxopentyl]amino}-5-[(2S,3R)-2,3,4-trihydroxybutyl]-3,5-dihydro-4H-imidazol-4-ylidene}-L-lysinate (DOGDIC), N^6^-{2-{[(4S)-4-ammonio-5-oxido-5-oxopentyl]amino}-5-[(2S)-2,3-dixydroxypropyl]3,5-dihydro-4H-imidazol-4-ylidene}-L-lysinate (DOPDIC), N^δ^-(5-methyl-4-imidazolon-2-yl)-L-ornithine (MG-H1), N^δ^-(45-hydro-5-(2,3,4-trihydroxybutyl)-4-imidazolon-2-yl]-L-ornithine (3DG-H1), N^δ^-(5-hydroxy-4,6-dimethylpyrimidine-2-yl)-L-ornithine (argpyrimidine), and 6-[2-[[(4S)-4-amino-5-hydroxy-5-oxopentyl]amino]-4-imidazo[4,5-b]pyridinyl]-L-norleucine (pentosidine). Modified protein surface models (light pink) are based on the structure of human hemoglobin (PDB ID 1COH).

**Figure 3 cells-11-01312-f003:**
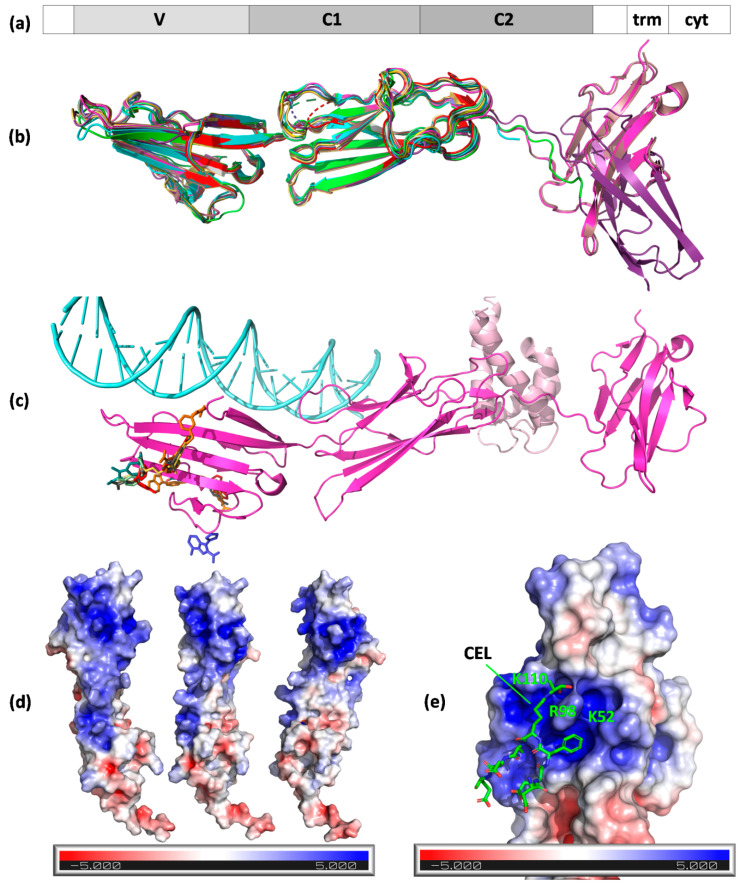
Structures of RAGE and ligand binding mode. (**a**) The domain organization of the receptor: 1–21, signal peptide; 23–116, Ig-like V-type; 124–221, Ig-like C2-type 1; 227–317, Ig-like C2-type 2; 343–363, transmembrane helical; 364–404, cytoplasmic (disordered); (**b**) A chains (N chain in the case of 3o3u, gold) of RAGE structure models aligned to 3cjj A chains (VC1 fragment, green). Protein chains are shown as cartoons and are colored: 4oi7, cyan; 4lp5, dark violet; 4p2y, dirty pink; 4ybh, pink; 6xq1, yellow; 6xq3, orange; 6xq5, olive; 6xq6, pale green; 6xq7, dark cyan; 6xq8, red; 6xq9, gray; and 7lmw, dark blue. MBP fusion protein in 3o3u is not shown. A chains (N in the case of 3o3u) of RAGE aligned to 3cjj A chain (VC1 fragment) with low or very low RMSD (given in Å): 3o3u–0.991, 4lp5–0.889, 4oi7–0.759, 4p2y–0.544, 4ybh–0.602, 6xq1–1.094, 6xq3–1.221, 6xq5–1.122, 6xq6–1.107, 6xq7–1.161, 6xq8–1.325, 6xq9–1.241, and 7lmw–1.264; (**c**) RAGE VC1C2 (cartoon, PDB ID 4p2y, magenta) with its ligands (S100-A6, cartoon, light pink, 4p2y; DNA, light cyan, 4oi7) and inhibitors (colored sticks, from PDB IDs 6xq–6xq9 and 7lmw); (**d**) positively charged patch on the surface of RAGE (surface view, PDB ID 3cjj) shown in three orientations; (**e**) CEL peptide binding to the positively charged patch on RAGE surface (PDB ID 2l7u).

**Figure 4 cells-11-01312-f004:**
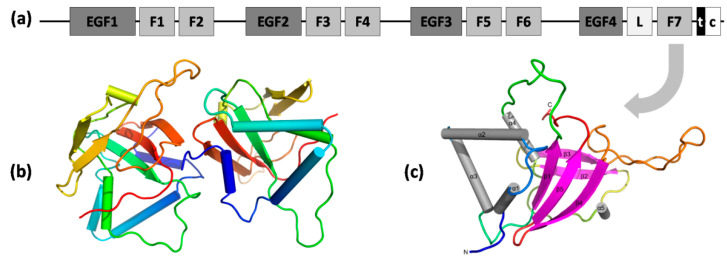
Domain organization of Stab1 and Stab2 receptors and the structure of FAS1 domain of Stab2. (**a**) Domains of stabilin receptors: EGF1–4—EGF-like domain repeats (in four series); F1–7—seven FAS1 domains; L—Link domain; t—transmembrane region; c—cytoplasmic (disordered) domain; (**b**) structure of the seventh FAS1 domain of Stab2 (PDB ID 5N86)—cartoon representation of chains A and B in an asymmetric unit (rainbow from blue N-terminus to red C-terminus); (**c**) different orientation of chain A with depicted secondary structure elements shown as on the cartoon model.

**Table 1 cells-11-01312-t001:** Classification of AGEs with respect to different criteria.

Source	Precursor	Chemical Structure and Ability to Emit Fluorescence	Molecular Weight	Physiological Importance
-Endogenous	-Glucose-derived (Glu-AGEs)	-Fluorescent and crosslinked	-Low (LMW AGEs)	-Non-toxic
-Exogenous (dietary, dAGEs)	-Fructose-derived (Fru-AGEs)	-Fluorescent and non-crosslinked		-Toxic (TAGEs)
	-Glycolaldehyde-derived (Glycol-AGEs)	-Non-fluorescent and crosslinked	-High (HMW AGEs)	
	-Glyceraldehyde-derived (Glycer-AGEs)	-Non-fluorescent and non-crosslinked		
	-Methylglyoxal-derived (MGO-AGEs)			
	-Glyoxal-derived (GO-AGEs)			
	-3-Deoxyglucosone-derived (3-DG-AGEs)			

**Table 2 cells-11-01312-t002:** The dietary AGE content in food.

Products	AGE ^1^ (kU/100 g)	Products	AGE (kU/100 g)
Meats		Carbohydrates	
Beef, raw	707	Bread, pita	53
Beef, steak, pan fried with olive oil	10,058	Bread, Greek, hard, toasted	607
Chicken, breast, skinless, raw	769	Biscuit (McDonald’s)	1470
Chicken, boiled with lemon	957	Cookie, biscotti, vanilla almond	3220
Chicken, breast, breaded, deep-fried, 20 min	9722	Donut, chocolate iced, crème filled	1803
Chicken, skin, back or thigh, roasted, barbequed	18,520	Apple, baked	43
Chicken, skin, thigh, roasted	11,149	Apple, Macintosh	13
Turkey, ground, raw	4957	Fig, dried	2663
Turkey, ground, grilled, crust	6351	Carrots, canned	10
Bacon, fried, 5 min, no added oil	91,577	Tomato	23
Bacon, microwaved, 2 slices, 3 min	9023	Vegetables, grilled (broccoli, carrots, celery)	226
Lamb, leg, raw	826	Honey	7
Lamb, leg, broiled, 450°F, 1 min	2431	Fats	
Fish		Almonds, roasted	6650
Salmon, fillet, microwaved	912	Butter, whipped	26,480
Salmon, fillet, broiled	3347	Margarine, tub	17,520
Trout, raw	783	Walnuts, roasted	7887
Trout, baked, 25 min	2138	Eggs	
Liquids		Egg white, large, 12 min	63
Milk, whole (4% fat)	5	Egg yolk, large, 12 min	1680
Juice, apple	2	Egg, omelet, pan, low, butter, 13 min	507

^1^ AGEs were assessed as carboxymethyllysine (CML) by enzyme-linked immunosorbent assay [31].

**Table 3 cells-11-01312-t003:** Human cell surface receptors for AGEs.

Receptor Type	Receptor Name (Other Names, Uniprot ID)	Ligands (Other than AGEs) or Receptor Function	Ref.
RAGE	RAGE (AGER; Q15109)	S100A12, S100B, amyloid-β peptide precursor (ABPP), oligonucleotides	[56]
Scavenger, class H	Stabilin-1 (Stab1, FEEL1; Q9NY15)	acetylated low-density lipoprotein (AcLDL), Gram-positive and Gram-negative bacteria	[57]
	Stabilin-2 (Stab2, FEEL2, HARE; Q8WWQ8)	hyaluronic acid (HA), heparin (Hep), chondroitin sulfate (CS), dermatan sulfate (DS), AcLDL, pro-collagen propeptides, oligonucleotides, phosphatidylserine (PS), bacteria	[57]
Scavenger, class A	SR-AI (P21757) and transcript variant SR-AII	modified low-density lipoproteins (LDLs)	[58]
Scavenger, class B	SR-BI (Q8WTV0), CD36	phospholipids, cholesterol (high-density lipoprotein; HDL), cholesterol ester, lipoproteins, PS	[59]
Scavenger, class E	OxLDL receptor 1 (LOX-1; P78380)	oxidatively modified low-density lipoprotein (oxLDL), HSP70 protein	[60]
AGE-Rs	AGE-R1 (OST-48/p60; P39656)	48 kDa subunit of the oligosaccharyl transferase (OST) complex	[61]
	AGE-R2 (80K-H/p90; P14314)	regulatory subunit β of glucosidase 2	[62]
	AGE-R3 (galectin-3; P17931)	galactose-specific lectin that binds IgE	[63]

**Table 4 cells-11-01312-t004:** X-ray structures of RAGE.

PDB ID	Title (Resolution, Space Group)	Protein Chain; Residues	Ligand	Ref.
3CJJ	Crystal structure of human RAGE ligand-binding domain (1.85 Å, P2_1_2_1_2)	A; 23–240	None	[68]
3O3U	Crystal Structure of Human RAGE (with MBP fusion) (1.50 Å, P2_1_2_1_2)	N; 23–231	None	[69]
4LP4/4LP5 (4OF5, 4OFV)	Crystal structure of the human RAGE VC1 fragment in space group P6_2_ (2.40 Å, P6_2_)/Crystal structure of the full-length human RAGE extracellular domain (VC1C2 frag.) (3.80 Å, P6_5_)	A, B; 23–231 /A, B; 23–323	None	[70]
4OI7 (4OI8 )	RAGE recognizes nucleic acids and promotes inflammatory responses to DNA (3.10 Å, P 6_1_)	A, B; 23–237	DNA (chains: E, F)	[71]
4P2Y (4YBH)	Crystal structure of the human RAGE ectodomain (fragment VC1C2) in complex with mouse (4P2Y)/human S100A6 (4YBH) (2.30/2.40 Å, I 2 2 2)	A; 23–323	Murine S100-A6 (chain B, 91 aa)/Human S100-A6 (chain B, 92 aa)	[72]
6XQ1–6XQ9, 7LMW	RAGE VC1 domain in complex with [inhibitor name] (1.51–2.50 Å, P 6_2_)	A, B; 20–231	Small molecule or fragment inhibitor ^1^	[73]

^1^ 6XQ1: 3-(3-(((3-(4-Carboxyphenoxy)benzyl)oxy)methyl)phenyl)-1H-indole-2-carboxylic acid; 6XQ3: 3-(3-{[3-(4-carboxyphenoxy)phenyl]methoxy}phenyl)-1H-indole-2-carboxylic acid; 6XQ5: Frag. 1 (7-methyl-3-phenyl-1H-indole-2-carboxylic acid); 6XQ6: Frag. 11 (3-phenoxyphenol); 6XQ7: Frag. 5 (5-bromo-3-methyl-1H-indole-2-carboxylic acid); 6XQ8: Frag. 1 and 11; 6XQ9: Frag. 1 and 13 (4-(3-hydroxyphenoxy)benzoicacid); 7LMW: 3-(3-((4-(4-carboxyphenoxy)benzyl)oxy)phenyl)-1H-indole-2-carboxylic acid.

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
