# Peer review of "Advanced Glycation End-Products (AGEs): Formation, Chemistry, Classification, Receptors, and Diseases Related to AGEs"

_cells, 2022, doi:10.3390/cells11081312_

Round 1

Reviewer 1 Report

The manuscript looks good overall. It covers the scope of the review very well.  However, the entire manuscript needs to careful proofreading and editing, as there are many grammatical errors that need to be corrected; grammatical errors appear in almost every sentence.  The following comments give some examples of these errors, but there may be others.

Examples of changes and revisions that need to be made:

  1. In lines 29 and 30 under section 1 (Introduction), can the phrase “on the way of non-enzymatic condensation” be clarified a little bit?
  2. The sentence in lines 37 to 40 under section 1 (Introduction) is not very clear. Can this be made clearer?
  3. The statement “the closer look was focused on the in vivo glycation of proteins” in lines 44-45 is not clear.
  4. Can the statement the “lower temperature of our organisms” be rephrased?
  5. Figure 1 on page 3 looks clumsy and a bit difficult to follow. Can it be broken down?
  6. In line 88, under Section 1.1, the statement “the compounds originated after Maillard and Polyol pathways” is not clear
  7. In line 486-487, it seems the word ‘ortientation’ needs to be checked for spelling
  8. The terminology needs to be checked. For example, in line 489 “binding place” may be better stated as “binding site
  9. In lines 626-627, the claim that diabetes mellitus “is considered a result of presence of AGEs in blood” needs to be checked for validation. Is it not more valid to say AGEs are the result of diabetes? Either way, this needs to be clarified and appropriate citations are needed.

Other comments and examples of errors:

Line(s)

Comment

3

Add comma between Receptors and and

9,10

Remove phrase in parenthesis

10

Remove comma after endogenously

13

Remove "the" prior to stable

15

Add comma after cardiovascular and remove "and" before neurodegenerative

19

Change required to require

22

Change exo- to exogeneous

29

Change "on the way" to "by means"

31

Remove "the"

32

Remove "The", capitalize reaction, and change it to reactions

32

Change "was already" to "have been"

33

Remove "ago"

34

Remove "e.g."

36

Add "the" before Maillard.

39

Rephrase sentence. Contaminant formation is not in line with increased safety or taste

44

Add comma after cancer

45-47

Shorten the sentence "Due to the lower…." to "In the human body, AGE formation is a prolonged process"

47

Add "observed" between first and endogenous

49

Add a period after [2] and add "This was" prior to followed

51

Change "of research" to publications

52

 Remove "the"

53

Change "rising" to "growth of"

64

Change "e.g." to "moieties like"

66

Could shorten sentence by removing mention of glucose and fuctose. If not, remove e.g. and replace with "like"

66

Change "the latter" to " Reducing ketoses"

67

Change "on" to "by"

70

Add "the" before Maillard.

78

Change "with" to "in"

109

Change "since" to although"

111

add "they are" before two separate groups

130

I might bullet point each item listed in the table (headers are oka how they are) as some entries span multiple lines

132

Add "defining and AGE" after criterion

136

Change beginning of sentence to "Endogenous AGEs can be furhter…"

137

Remove "e.g"

138-141

I think there may be excess spacing in some compounds

150

I might only refence figure 2 here.

137-150

This list is difficult to read in a paragraph. I might mention the class and maybe some important examples, but refer to figure 2 as the main listing

151-168

For readability, I might move the names from the figure caption and place each name below the abbreviation and structure.

169

Change "an impact" to "impacts"

170

Change "or" to "and"

170

Should "in this case" be replaced with "in the case of long-lived proteins and peptides"

171

Change "such situations" to "this often" and remove "mostly"

172

Remove "observed"

197

Change "favoring" to "favorable for", make AGE singular

211

Use full name for TU

235-237

Change sentence to "Further methods for the determination of AGE levels, wether or not they rely on intrensic fluorescence, will undoutably be developed as novel biomarkers."

240

Remove sentence "it should be noted however…."

241

Remove "And so,"

241

Change "which" to "whose"

245

Change "the fact of" to "their"

247

Remove "however"

249

I might change "is" to "can be"

251

Change "it" to "this"

252

Add a period behind body, remove "and", capitalize "their", and add "the" prior to liver

255

Remove "e.g. exact", add comma behind diet

257

Remove "very"

261

Change "i.e." to "such as"

262

 add "and" before gold and remove ",etc"

276

Add "the" before Takeuchi

279

Run on sentence. Replace ", e.g." with ". Examples include"

279

Change "/" to ","

288

Change "which" to "whose"

288

"violates" is an odd choise. Something like "alters" may be better suited here

292

Remove "since years"

292

Remove comma before such.

301

Add "the" prior to authors

328

comma instead of semicolon

330

Add a period after [61] and capitalize "the"

331

Remove "On the" and capitalize "contrary"

332

Change "ligand's" to "ligand"

337

Change "already gave an" to "provided and change "of" to "for"

340

End entence at "Sirosis et al.", remove "whereas" and start new sentence with "The model…"

342

Remove "very recently"

355

Should "chains" be singular?

356

Change "present" to "presents" and change "chains" to "chain"

357

Specify which receptor

358-359

Remove "that overlaps slightly with dna binding site"

359

Remove , after parentheses

361-375

Should this have a literature citation?

377-378

The lys and arg positions can probably be removed here

380

Remove "possibly"

389

Add "when" prior to "analysing"

391

Add "The" prior to "DNA"

395

Change "ie." to "i.e.". Add "and" before 48 and 52

397

Move "simultaneously" before "bind"

400

Remove "the"

404

Add comma before "and amphoterin"

404

Start this sentence as "This signaling cascade triggers the activation of the ERK1/2 and p53 pathways…"

405

Run on sentence. Add a period after (MAPK) and replace "e.g. by" with "For example,"

406-407

Rewrite this sentence for clarity.

409

Remove comma

416

Change to …"endothelial cells. This includes the elimination"…

418

Replace "the same group" with the name of the group.

420

Change "endothelium" to "endothelia" and remove the comma following it

421

I think "ad" should be "and". Rather than using FEEL1 and 2, I think Stab should be used. Consistent nomenclature will help clarity.

427

"mass" should be "masses" and add "the" prior to "lighter"

435

Change to …"act as scavenging receptors"…

439

Change "FEEL1" to "Stab1"

452

Add a comma after "RHAMM"

452

Remove "190 kDa and 315 kDa"

457

Add a comma after "region"

459

Change the comma between "periostin" and "Big-h3" to "and"

489

Change "AGEs" to "AGEs'"

490

Change colon to period behind "unknown" and add "This raises" prior to "the question"

492

Remove "of" and "remains open"

494

Remove "neither"

498

Remove "moreover" and capitalize "despite"

503

Remove "and", add a period behind GAGs, and capitalize "the"

504

"or" could be changed to "and"

508

Add comma behind "SR-AI" and remove comma after "SR-AII"

519

Change "and" to a comma

524

Change "at" to "in"

525

Change "and" to a comma

526

Remove "at", Replace "where" with a period, and capitalize "the"

533

Add a comma after "chaperone"

534

Add "the" after "of" and remove "in"

537

Remove comma

539

Remove comma

540

Remove "moreover"

543

After "available", change comma to a period and capitalize "However"

543

Add comma after "out of which"

544

Remove "the"

545

Add a period after parenthesis, remove " and one", and capitalize "the"

546

Change hyphen to "is" and add closing parentesis after "3VLG"

553

Add period after reticulum and change "that" to "OST"

556

Add "the" before "DDOST" and remove "an"

557

Add "the" prior to "endoplasmic"

558

change to ..."that sequentially cleaves the two"...

559

make "glycoproteins" plural

561

Change "which" to "that"

562

Add "the" before "cytoplasm"

566

Change "were" to "have been"

568

Change "whereas" to "however"

569

Change to …"increases AGE removal"…

571

Change "receptor's" to "AGE-R1's"

572

Remove "the"

573

Remove comma

574

Change "receptor's" to "AGE-R1's", change colon to period, and capitalize "studies"

579

Add "a" between "had" and "95%". Use OST acronym.

580

Change beginning of sentence to "This OST-48 N-terminal"...

581

Remove "that"

582

"identified as an 80-90"

586

End sentence after [102] and start a new one

588

Remove comma after "plaques"

591

..."further increasing AGE-ligand"...

592

Add "a" before "high"

598

Change colon to a period and capitalize "after"

600

Remove both "the"s and the comma

603

Add a comma after "proteinuria"

606

Change colon to start of a new sentence

607

Add comma after "RAGE"

612

change "its" to "glucose" and add comma after "tissues"

613-617

Run on sentence

618-619

Change to …"as collagen, further extending the life of the protein"

619

Add "the" before "covalent"

621

Make barrier plural

626

Change "popular" to "common"

627

Add "the" before "presence, delete "also", and capitalize "other"

629

Viral infections seem out of place here. Viral infections may be made worse by the presence of AGEs but they are not a cause

629

Add a comma after "problems"

630

Change to "indirect effects of AGEs."

630

"there is evidence"

631

Remove "of a kind", replace hyphen with "where"

632

Remove "additionally"

635

Change "is" to "are"

636

Add "the" before "gastric"

638

"where they are"

640

Remove the parentesis and the words inside it

641

Trypsin

644

Change "on the" to "with"

644-645

"The level of absorption is different for various AGE structures and is particularly dependent on molecular weight."

649

"hypothetical"

650

comma after "paracellular"

651

Remove "the"  before "simple" and remove "way"

652-654

Im not sure what is meant by this sentence.

658

Comma after "digestion"

660

General note: I would avoid using "Ca.". I think approximately is a better choice. Change hyphens to commas

661

Remove a

662

Change "is" to "are" and swap "lower" to before "intestinal"

662-663

Change to "These microbiota have different enzymatic abilities than humans and degrade AGEs in different ways (e.g. via fermentation)."

667

General note: Don't say "proved". Better choices include suggest, demonstrate, or concluded

667

..."that high levels of"…

669

Add a comma before and

675

Add a comma before and

676

Use of Ca. and should the range be 5-8?

687

Change tittle to just "Excretion"

688

Change "is" to "are"

688-689

Change to ..."while in individuals with diseases, such as duabetes mellitus, it is"...

691

"the lower the ratio"

691

Change "ability" to "rate"

692

Remove "the"

695-696

This statement contradicts findings above (689). Is urinary excretion constant? Is the literature in disagreement? Is it a difference in humans vs cats and dogs?

693

"A study from"…

695

Change "on" to "with"

697

"the level of"

698

Change proves

700

"experiment were complimentary with human studies, work on animal models"…

701

"differences in their gastointesinal tract"

703

Incomparable seems like a strong word choice. Imprefect or inconsestent may be better as sometimes comparisons may work well

703

Cite and/or name the initiative mentioned

704

Add "models" after "proposed"

707

Add comma after enzymes

723

Use of proven. Change to "demonstrated"

728

Add "the" after "that"

731

Add "the" after "when"

739

"oxidents" may be a better alternative to "ROS" to also include "RNSs"

740

"induce"

743

"A comprehensive study"

747

Change colon to period

748

"induce a pathway" or "induce the pathway"

750

Add "the" before "proinflammatory". Change hyphen to period. Capitalize "Biochemical"

751

..."tissues associated with aging are often"...

754

 Comma before "and"

755

…"results in the production of the pro-inflammatory effector"…

756-757

…"(C5a),increases other inflammatory markers such as IL6,IL-1a,ICAM, TIMP-1, and leads to"…

758-759

I'm unsure what this sentence is conveying

771

Comma before "and"

773

Comma after "cells"

774

"cannot be directly compared"

774-776

"One of the difficulties of measuring changes in endogenous AGEs in the human body is the naturally high background levels."

777

Remove "as the most common AGE," and remove "the"

779

Remove "immunologic method" and parenthesis around ELISA

781

Remove "way"

782-783

 "For example, after the intake of food, the pH of the stomach is approximately 1-1.5."

783-784

"During digestion, pH increases to 5-7. After about 3 hours, pH returns to the initial range of 1-1.5"

782-784

Specify the stomach somewhere in here.

784-785

"In many studies, the pH is averaged to 2-2.5 for approximately 2 hours."

785

What is the TIM-1 model?

787

..."(SAF) allows the detection of"...

788

Move diabetic comorbidities to end of sentence as this descrbes all of these conditions

789

Comma before "or"

788

Is comorbitidies the right choice of words? Should it be complications of diabetes?

791

"measuring"

792

Remove colons and semicolons. Use commas here.

793

Remove "method"

794

I would only mention AGE reader, especially if SCOUT was removed due to technical problems and poor results.

795

Move "(DiagnOptics)" to first mention of AGE Reader

797-798

"The score is expressed as the ratio of light"…

799

Remove "the" after and.

803

"nephropathy, bone strenth weakness [170], neuropathy [171],"…

804

Change "for" to "a"

805-806

..."state. AGEs are a possible factor for causing diabetes states and are believed"...

807

…"carriers, creating a"…

807

Add comma after "etiology"

810-811

"which are responsible for recognizing and"

811

Add "the" after "on". Change "proven" to "demonstrated"

816

Change "proven" to "known"

817

...",which are responsible"…

818

Change "on" to "to determine"

820

Change "or" to "and"

822

Change "dedicated" to "significant"

823

Remove "also the"

824

Change "proved" to "suggested"

825

"CVDs", add a comma after "CeVD"

833

Add a comma before "and" and "the" after inducting

838

Clarify sentence

839

..."amounts of methionine sulfoxide increases the risk of CVD. On the other hand"...

841

"tested the correlation"

842

comma after levels

845

Change "examined" to "likely"

846

Changed "proven" to "found"

847

"non-diabetic postmenopausal Indian women"

848-849

…"with non-diabetic patients"…

849

Move citation to end of sentence

851

…"b-thalassemia, the most severe form of CAD,"…

854

Remove "soluble form of"

855

"Being a vasculopathy"

856

"sRAGE levels are significantly increased"

861-864

Reword for clarity.

865

"creating oxidative stress"

866

Start a new sentence at "Therefore,"

868

"have been confirmed"

868

Change hyphen to comma

869

Change "is" to "are"

870-871

"Estrogen is an antioxident and studies show"…

874

"This increases"

876

Remove "the" after "up-regulating"

880

Remove both "the"s

888

Change "confirms" to "suggests"

890

"Also,"…

893

Change "the" to "an"

910

Change "proved" to "demonstrated"

911

"kidney diseases. RAGE knockout"

915

Comma before "or"

917

"Bunches of the"…

917-918

Change hyphens to commas

920

Remove "the" before "oxidation"

921

Comma before "or"

921

"Damage to these tissues are the result of"…

934

Remove "the"

939

"inflamation, hypoxia, and oxidative stress are microenvironmental"…

940

"These three states"…

942

"chronic diseases, and obesity, are also"….

950

Comma before "and"

956

Comma before "or" and after "also". Change "model" to plural.

957

"exposed higher levels of" and change "group" to plural

958

"tumor promotion was also studied in mouse models of"

966

"Two cell lines of colorectal cancer, with or without Glu-AGEs, were chosen for examination."

968

move "respectively" after "MMP-2"

969

Comma before "and"

971

"from an organism"

972

"Kupher cells. This results"

973

Add "the" before "liver"

974-977

Run-on sentence

982

Remove "the"

984

Correct citation style.

989

"Infertility is listed as a complication of diabetes mellitus complication, however,"...

991

Swap "women" and "men" so the former and latter statements make sense in 991 and 992

991

…"structure. In the former"…

992

Remove "the" before "ovarian"

992

Change hyphen to comma

994

Comma before "and"

994-995

"aging. The diagnostic criteria accompanying"…

996

Comma before "or"

997

"Increased amounts of AGEs cause oxidative"….

998

"change" instead of "changes"

999-1000

…"the effects of AGEs are not limited to enhancing oxidative stress because"…

1002

Change "proven" to "demonstrated"

1005

Remove "the"

1006

"that the polyvalent"

1014

Change "proven" to "shown"

1016-1017

..."statistical relationships have been found between the course of the pregnency and live-birth ratio with AGE levels"...

1023-1024

Correct citation style.

1025-1026

diabetes, hyperglycemia, hypertension, heart disease, and obesity.

1033

"detecting" to "determining"

1034-1035

General note. Be consistent in naming the disease. Use Covid-19 or SARS-CoV-2 (unless simply mentioning its also called the one you don’t use)

1034

Change "disease" to "patients"

1035

Add a comma after "cRAGE" and change "is" to "are"

1038

Comma before "and"

1041

Change "even though" to "As"

1045

"with receptors is limited"

1048

"preventing elevated AGE levels  which cause pathologic"

1051

Extra comma

1052

Remove parentesis and "as reviewed by"

1053

Should "AGE" be "dAGE"

1064-1065

I'm not sure this is needed for the acknowledgements.

Reviewer 2 Report

The presented review is devoted to the analysis of data related to advanced glycation end-products (AGEs). AGEs aroused the interest of the scientific community due to the increasing evidence of their involvement in many pathophysiological processes and diseases that have become widespread in recent times,  such as diabetes, cancer, cardiovascular and neurodegenerative diseases.  The review is distinguished by the breadth of coverage of the problem, it considers the formation, chemistry, classification, reception, intracellular degradation, secretion of AGEs and also diseases related to AGEs. This makes the review interesting for scientists of different scientific fields. The review is logically structured and based on the analysis of a large amount of literature data.

I have a few minor remarks.

  1. Abbreviations

The review uses a lot of abbreviations, which in some cases makes the text difficult to understand. Of course, one cannot do without abbreviations when naming chemical compounds, but the names of diseases can be written in full.

Line 756  the abbreviation C5a is not used anywhere in the following text

Line  859 the abbreviation ALS is not used anywhere in the following text

Lines 1030 and 1033:  Is it necessary to indicate abbreviations COPD, HFNO and MV?

The authors could provide a list of abbreviations for the convenience of readers.

  1. Citations

Line 257 “Takeuchi et al. has very recently proposed …” reference is not provided

Line 269 “Takeuchi et al. points at a specific…”  reference is not provided

Line 276 “Many publications from Takeuchi group…” Links to these publications are not provided

Line 285 “Takeuchi et al. establishes a strong…” reference is not provided

Line 290 “(WHO) in 2015 recommended the reduction of daily sugar intake…” reference is not provided

Line 1023 (Rojas, Schneider, Lindner, Gonzalez & Morales, 2021; Stilhano et al., 2020) – wrong link formatted

Line 1688 Gonza`lez, I.; is it correct?

Please, close the parenthesis in Line 1050

Line 281 “excessive cellular accumulation of TAGEs can lead to their damage or death…”  can lead to damage of cells? Not TAGEs damage?

Round 2

Reviewer 1 Report

The authors have addressed my prior concerns